# Economic interests cloud hazard reductions in the European regulation of substances of very high concern

Jessica Coria [1] ✉, Erik Kristiansson[2] & Mikael Gustavsson[1,2]

Here we investigate how the conflicts between hazard reduction and economic interests have shaped the regulation of substances of very high concern (SVHCs) under the Authorization program of the European chemical regulation Registration, Evaluation, Authorization, and Restriction of Chemicals (REACH). Since regulation starts with listing SVHCs on the Candidate List, we analyze the relative importance of toxicological properties, economic motivations, and available scientific knowledge on the probability of inclusion on the Candidate List. We find that the most important factor in whether a substance is listed is whether it is being produced in, or imported into, the European Economic Area (EEA), with the regulators less likely to place a substance on the list if it is currently being produced or imported in the EEA. This evidence suggests that regulators have listed chemicals of secondary importance leading to lesser than anticipated hazard reductions, either because production and imports had already ceased before the listing, or because the compound has never been produced or imported in the EEA at all.

Estimates by the European Environment Agency suggest that 62 percent of the volume of chemicals consumed in Europe in 2016 were hazardous to human health, with the potential to cause a range of diseases, including cancer; fetal malformations; diseases of the respiratory, endocrine, cardiovascular and urinary systems; and neurodevelopmental and immune disorders[1]. The same year, the World Health Organization estimated that the burden of disease due to chemical exposures accounted for 1.6 million lives and 45 million disability-adjusted life-years lost[2]. These WHO estimates are based on a selection of chemicals with sufficient evidence for global quantification of health impacts. However, people are exposed to thousands of chemicals from a wide range of sources, many of which have not been evaluated for their potential health and environmental effects[3–5].

In the European Union, the use and production of chemicals are covered by several different regulations, including Registration, Evaluation, Authorization, and Restriction of Chemicals (REACH). REACH addresses chemicals used in industrial processes and intentional chemical mixtures and chemicals added to products in the European Union[6,7]. Thus, REACH does not address substances covered by more specific regulations (such as medicines and agricultural chemicals). As of February 2020, 22,425 unique substances had been registered under REACH.

REACH consists of four complementary programs[7,8]. Through the Registration program, companies are required to submit dossiers containing information about the properties and uses of chemicals. Through the Evaluation program, the European Chemical Agency (ECHA) checks some of these dossiers for compliance with the information requirements. The two key REACH programs that regulate chemical risk are Authorization and Restriction. Through the Authorization program, the manufacture and use of substances of very high concern (SVHCs), which may have serious effects on human health and the environment, can be subjected to binding limitations and conditions, including complete prohibitions. The Restriction program also seeks to ensure that the risks from hazardous substances are properly controlled by prohibiting individual problematic uses of specific substances. However, unlike Authorization, Restriction is not limited to SVHCs. This study investigates the listing of SVHCs under the Authorization program.

[1]Department of Economics, University of Gothenburg, Gothenburg, Sweden. [2]Department of Mathematical Sciences, Chalmers University of Technology/University of Gothenburg, Gothenburg, Sweden. ✉e-mail: Jessica.Coria@economics.gu.se

The main aims of REACH are to ensure a high level of protection for human health and the environment, including the promotion of alternative test methods, as well as the free circulation of substances on the internal market and the enhancement of competitiveness and innovation[7–9]. Criteria for identifying substances as SVHCs are whether they are carcinogenic, mutagenic, or toxic for reproduction (CMR); persistent, bioaccumulative and toxic (PBT); very persistent and very bioaccumulative (vPvB); or raise equivalent levels of concern[9]. CMR substances are directly hazardous to human health, while PBT and vPvB substances pose long-term, unpredictable risks due to their longevity, irreversible nature, and tendency to accumulate in the food chain.

The route to authorization starts when a member state or the European Chemical Agency, at the request of the European Commission (EC), proposes substances for inclusion on the Candidate List of SVHCs. The substances included on the Candidate List then undergo a prioritization process to be included on the Authorization List. In turn, substances placed on the Authorization List cannot be made available on the market after a defined sunset date, unless the EC grants an authorization. Inclusion of a substance on the Candidate List is thus a first step toward requiring EC authorization for a compound's manufacture, import and use[10].

If a member state or the EC wants to propose a substance for the Candidate List, they need to submit a dossier with information about the toxicological properties of the substance. Once submitted, a 45-day consultation period starts, during which anyone can comment and provide additional information on the substance's properties, uses, and available alternatives. A substance is listed as a SVHC if no objection to the listing is made, or if a committee of national representatives agrees on the listing after considering information submitted during the public consultation[11].

Substances on the Candidate List are then prioritized for inclusion on the Authorization List by the European Chemical Agency. Priority is given to substances with persistent, bioaccumulative, and toxic properties, widely dispersed use, or high volumes of production or use. The committee of national representatives provides a recommendation for inclusion on the Authorization List considering the information received during the Candidate List consultation and through a second round of public consultations. Finally, the EC decides which substances to include on the Authorization List based on that committee's recommendation.

As of February 2020, 303 substances were listed as SVHCs on the Candidate List, and 86 of those were on the Authorization List. In contrast, in 2013 the European Chemical Agency predicted that there should be 1,500 SVHCs addressed by REACH[12]. To date, most of the chemicals that have been suggested to the Candidate List have been listed (i.e., only about 6% of the dossiers submitted have been withdrawn or the substances not identified as SVHCs). Moreover, the committee of national representatives has mostly unanimously agreed on the identification of SVHCs (i.e., only about 4% of the dossiers submitted were resolved without full agreement). However, the low number of SVHC substances listed so far raises concerns about whether the listing procedures can adequately and timely control the risks posed by SVHC. It also raises the question of whether the Candidate List is being shaped by those interest groups that are more successful in translating their preferences into policy outcomes[13,14].

Empirical evidence shows, for instance, that business and industry interests are considerably better represented in public consultation processes than environmental and consumer interests[15,16]. Furthermore, organizations representing diffuse interests, such as environmental NGOs and local authorities, perform significantly worse in achieving their preferences than main business groups that represent concentrated interests[17,18]. The structure and economic importance of the domestic chemicals industry might also be of considerable importance for the listing of SVHCs. Specifically, chemicals produced or used in large quantities, or by many EU countries, might be more difficult to regulate because they affect the economic interests of many actors, and the likelihood of successfully lobbying policymakers increases with the number of interest groups pushing for the same policy goal[19,20]. The available scientific evidence about the detrimental effects of the chemicals might thus be challenged by companies, succeeding in diverting attention from their own products.

A related problem is uncertainty about the negative effects of chemicals, which can also lead to a legislative standstill[21]. There has often been a lengthy process between the first scientific early warnings and the subsequent policy action[22]. Policymakers have been more likely not to regulate something that was later found to be harmful than to err on the side of caution[23]. The REACH listing procedures of SVHCs might enhance this tendency, as the consultation process provides producers with an opportunity for exposing weaknesses and uncertainties in the scientific case for the inclusion of chemical substances on the Candidate List[24].

In this paper, we investigate the relative importance of toxicological properties, economic motivations, and available scientific knowledge as drivers of the listing of SVHCs on the Candidate List. Our choice of variables is based on the criteria for identifying substances as SVHCs, and on a review of relevant literature. First, since SVHCs are explicitly defined based on their toxicological properties, we investigate whether the toxicity of a chemical substance is the main determinant of inclusion on the Candidate List. Second, we suspect that chemicals produced or used in large quantities or by many EU countries will be more difficult to regulate. We believe this is the case because of extensive documentation that it is politically difficult to implement policies that affect the economic interests of many stakeholders, and that industrial stakeholders try to shape policies to reduce harm to the industry (see e.g.[25,26]). Third, since it is well known that scientific evidence provides decisionmakers with political support for policy implementation[6,21], we suspect that widely studied chemicals are easier to regulate.

We operationalize these variables by means of indexes of toxicological properties, tonnage, and number of countries with ongoing or previous production/import within the European Economic Area, and an index accounting for the availability of scientific studies analyzing the effects of the chemicals. Since the relative importance of these factors is affected by the sample of chemicals used to perform our analysis, we compare the Candidate List to three alternative lists of chemicals. The first is all 22,425 chemicals registered under REACH as of February 2020. The second is the SIN (Substitute It Now) list of hazardous substances developed by the International Chemical Secretariat, ChemSec (an independent non-profit organization that advocates for substitution of toxic chemicals). The third is a list of hazardous substances that should be phased out, developed by the Swedish Chemical Agency (PRIO list). Both the SIN and PRIO lists have the explicit aim of raising awareness about chemicals that qualify as substances of very high concern according to the criteria specified by the REACH regulation. However, in contrast to the Candidate List, which is the result of a participatory process in which different stakeholders can shape the outcome of the list, the SIN and PRIO Lists are elaborated by experts, and, hence, less prone to political roadblocks. As of February 2020, the SIN list included 999 chemicals, while 1938 substances were listed on the PRIO list. Finally, we also compare the Candidate List to the Authorization List, which, as described above, includes the subset of substances on the Candidate List, subject to binding limitations and conditions.

## Results
### What drives inclusion on the Candidate List?
We investigate and compare the CMR and eco-toxicological properties of substances included on the different lists—REACH, Candidate, Authorization, SIN, and PRIO—by means of the CMR Score and

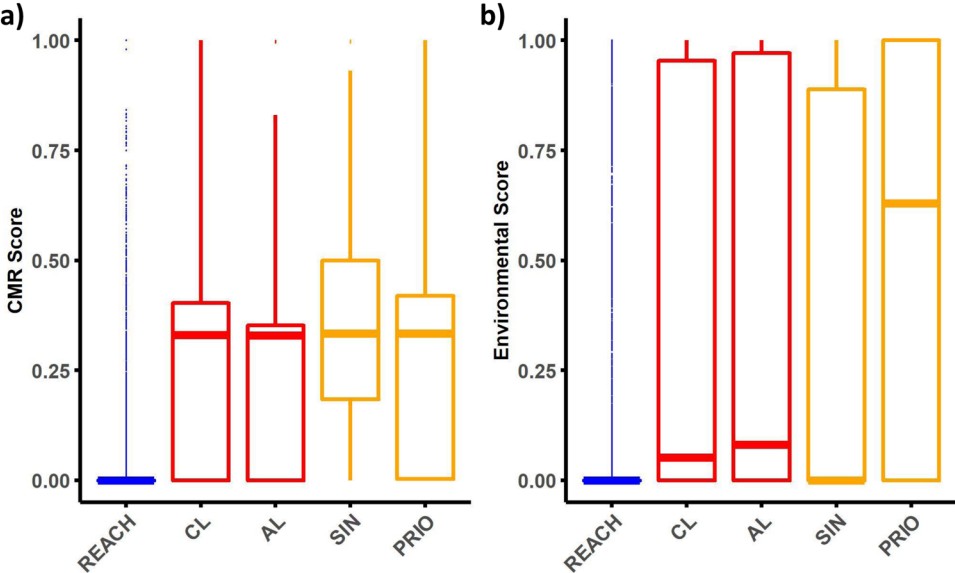

**Fig. 1 | Distribution of CMR and Environmental Score across chemical lists.** Panels (**a**) and (**b**) present the distribution of the CMR Score, measuring Carcinogenicity (C), Mutagenicity (M), and Reproductive toxicity (R), and the Environmental Score, measuring environmental hazard, across the five examined chemical lists. Note that some compounds could not be assigned a score as there was no matching entry in the classification and labeling inventory. Sample size (**a**):

European chemical regulation Registration, Evaluation, Authorization, and Restriction of Chemicals (REACH, $n = 16,846$), Candidate List (CL, $n = 247$), Authorization List (AL, $n = 79$), SIN List (SIN, $n = 877$), PRIO List (PRIO, $n = 848$). The parallel bars represent the median value while the hinges show the first and third quartiles. The whisker extends 1.5 inter-quartile ranges from the hinge. Outliers are plotted as individual points.

Environmental Score. These scores are based on the percentage of products that are classified as having CMR properties and toxic effects on aquatic life in accordance with the Globally Harmonized System (GHS). The CMR score is higher for substances listed in the Candidate List than for the bulk of substances registered in REACH (Fig. 1a). In contrast, the distribution of the CMR score is quite similar between the substances listed on the Candidate List and the Authorization List and between the Candidate List and PRIO, while the distribution under the SIN list seems more skewed toward higher values of the CMR score. The mean CMR score of the substances on the Candidate List is significantly higher than that of the substances in the REACH registration, and lower than that of the substances on the SIN List, while there are no statistically significant differences between the Candidate List and the Authorization List or between the Candidate List and PRIO (Table 1). The description above also holds for the Environmental Score, except that it is the distribution of substances listed under PRIO which seems skewed toward higher values of the Environmental Score (Fig. 1b). This is also true for the mean value of the Environmental Score under PRIO, which is significantly larger than that of the Candidate List (Table 1).

Overall, the comparison indicates that, from a CMR perspective, the substances listed under SIN are significantly more hazardous than the substances on the Candidate List; from an environmental perspective, the hazardousness of substances on PRIO is statistically larger than those on the Candidate List. Moreover, we observe no significant differences between the CMR Score and the Environmental Score of substances listed on the Candidate and Authorization Lists (Table 1).

There are also significant differences among the lists when it comes to economic and knowledge parameters (Table 1). Compared to the SIN list, the substances on the Candidate List are produced in lower volumes and by fewer countries. The opposite holds when comparing PRIO to the Candidate List.

We proxy for the scientific knowledge available by means of a publication rank that varies between zero and four, where zero denotes a compound with no information and four denotes a very well-studied compound (see Methods). The publication rank is significantly higher for substances included on the Candidate List as compared to

SIN and PRIO (Table 1). This finding supports the view that it is easier to regulate chemicals for which there are well-documented effects.

Finally, when it comes to the comparison of the substances on the Authorization vs. the Candidate List, we observe no significant difference among the economic or knowledge parameters.

**Toxic hazards versus economic interests and the candidate list**
We used logistic regressions to analyze the relative importance of toxicity, economics, and available scientific knowledge on the probability of inclusion on the Candidate List, and the differences between the Candidate List and REACH, SIN, and PRIO. We also verified whether our findings hold in the analysis of the probability of inclusion on the Authorization List. All variables were normalized to range between 0 and 1 (see Methods).

Figure 2 plots the marginal contribution and the corresponding confidence intervals of each variable. A marginal contribution larger than zero indicates a positive association between the driver and the probability of inclusion on the Candidate List, while there is a negative association when the marginal contribution is smaller than zero.

When it comes to the sample of all substances registered under REACH, major drivers of inclusion of a specific compound on the Candidate List are the CMR Score, the number of countries producing or importing the substance, and the amount of available scientific knowledge (Fig. 2a). If the marginal contribution of a CMR Score is equal to one, the odds of being included on the Candidate List are 184 times higher than the contribution of a CMR Score equal to zero. By analogy, if all countries in the European Economic Area were producing or importing the substance, the odds of inclusion would decrease 520-fold, compared to the case if no country were producing or importing the substance. Finally, a lack of publications on the substance reduces the odds of inclusion on the Candidate List by a factor of 46, compared to substances that are very well studied.

In turn, the quantity produced/imported (tonnage band) and the number of countries producing or importing the substance are the major drivers explaining which of the substances included on the SIN list are also listed on the Candidate List (Fig. 2b). In contrast, CMR properties have a larger and positive effect in explaining inclusion on

**Table 1 | Mean comparisons across lists of chemical substances**

| Mean | REACH (1) | CL (2) | AL (3) | SIN (4) | PRIO (5) | REACH-CL (6) | AL_CL (7) | SIN_CL (8) | PRIO_CL (9) |
|---|---|---|---|---|---|---|---|---|---|
| *Toxicological parameters* | | | | | | | | | |
| CMR Score | 0.03 (0.10) | 0.29 (0.26) | 0.27 (0.27) | 0.37 (0.26) | 0.32 (0.26) | 2.5E−131*** | 4.6E−01 | 4.4E−06*** | 3.1E−01 |
| Carcinogenicity Warning | 0.02 (0.12) | 0.07 (0.44) | 0.04 (0.46) | 0.05 (0.46) | 0.06 (0.48) | 7.8E−09*** | 1.8E−01 | 4.6E−02** | 3.2E−01 |
| Carcinogenicity Danger | 0.03 (0.15) | 0.28 (0.44) | 0.32 (0.46) | 0.55 (0.46) | 0.42 (0.48) | 1.4E−80*** | 3.7E−01 | 8.9E−22*** | 1.6E−06*** |
| Mutagenicity Warning | 0.03 (0.14) | 0.18 (0.37) | 0.15 (0.34) | 0.30 (0.42) | 0.21 (0.39) | 1.3E−21*** | 2.1E−01 | 3.4E−02** | 1.2E−02** |
| Mutagenicity Danger | 0.01 (0.09) | 0.09 (0.27) | 0.08 (0.27) | 0.16 (0.33) | 0.05 (0.22) | 6.3E−22*** | 9.3E−01 | 2.5E−04*** | 9.8E−03** |
| Reproduction Warning | 0.02 (0.14) | 0.12 (0.30) | 0.09 (0.24) | 0.12 (0.26) | 0.06 (0.21) | 2.3E−20*** | 2.4E−01 | 9.9E−01 | 7.2E−06*** |
| Reproduction Danger | 0.02 (0.12) | 0.37 (0.47) | 0.33 (0.46) | 0.25 (0.41) | 0.34 (0.46) | 6.3E−139*** | 3.5E−01 | 2.0E−08*** | 1.1E−01 |
| Environmental Score | 0.08 (0.24) | 0.40 (0.45) | 0.40 (0.44) | 0.31 (0.43) | 0.51 (0.47) | 2.0E−62*** | 8.8E−01 | 6.2E−04*** | 3.9E−05*** |
| Acute Environmental Warning | 0.09 (0.26) | 0.40 (0.46) | 0.43 (0.47) | 0.31 (0.44) | 0.51 (0.48) | 6.8E−56*** | 5.1E−01 | 3.0E−04*** | 3.1E−04*** |
| Chronic Environmental Warning | 0.08 (0.25) | 0.39 (0.46) | 0.37 (0.46) | 0.31 (0.43) | 0.52 (0.48) | 7.3E−59*** | 7.0E−01 | 2.0E−03** | 9.8E−06*** |
| *Economic parameters* | | | | | | | | | |
| Tonnage Band | 1.56 (1.69) | 1.51 (2.00) | 1.41 (1.91) | 2.33 (2.67) | 0.40 (1.24) | 6.0E−01 | 5.6E−01 | 2.9E−07*** | 5.0E−41*** |
| #Countries[a] | 1.36 (0.56) | 1.04 (1.36) | 0.81 (1.29) | 1.53 (1.65) | 0.10 (0.65) | 7.4E−04*** | 2.6E−01 | 2.3E−02** | 3.0E−43*** |
| *Knowledge parameters* | | | | | | | | | |
| Publication Rank | 0.22 (0.58) | 0.87 (1.07) | 0.82 (1.02) | 0.67 (1.02) | 0.36 (0.80) | 2.0E−63*** | 5.5E−01 | 4.0E−06*** | 4.6E−29*** |

Cols (1)–(5) report the mean value and standard deviation (in parentheses) of the toxicological, economic, and knowledge parameters.
Cols (6)–(9) report the *p*-value and statistical significance of a two-way t-test comparing the mean value of the parameters of interest for the chemicals included in the Candidate List (CL) to those listed under the European chemical regulation Registration, Evaluation, Authorization, and Restriction of Chemicals (REACH), Authorization List (AL), SIN List (SIN), and PRIO List (PRIO), respectively.
The CMR Score measures Carcinogenicity (C), Mutagenicity (M), and Reproductive toxicity (R), and the Environmental Score measures environmental hazard.
***Significant at 0.01, **Significant at 0.05, *Significant at 0.1.
[a]Mean and standard deviation for the number of registered countries were based on the variance-stabilizing transformed data.

the SIN list. This indicates that, in relative terms, the SIN list seems to have a greater focus on CMR properties than the Candidate List.

Figure 2c shows that tonnage band and the Environmental Score are the major factors explaining which of the substances included in the PRIO list are also listed on the Candidate List. In comparison to the substances listed on the PRIO list, there are no statistically significant effects of CMR properties in explaining the probability of inclusion on the Candidate List. In contrast, the negative and statistically significant coefficient for the Environmental Score suggests that the PRIO list has a greater focus on environmental properties, compared to the Candidate List.

All in all, these results indicate that the Candidate List has successfully included some hazardous chemicals produced or used in Europe. However, the substances listed on the Candidate List are no more hazardous than the substances listed on alternative lists of hazardous substances, either in terms of CMR properties or detrimental environmental effects.

The economic variables under study are significant in all samples. Nevertheless, whether they have a positive or negative effect on inclusion on the Candidate List depends–again–on the sample of chemicals at hand. Compared to REACH, inclusion on the Candidate List is most likely for those substances with higher tonnage produced or imported by fewer countries. Compared to the SIN list, substances on the Candidate List are produced or imported in lower quantities but by a larger number of countries. Compared to the PRIO list, substances on the Candidate List are produced in higher quantities, with no significant differences in terms of the number of countries.

The positive effects of the number of publications in explaining inclusion on the Candidate List are robust across all our samples. However, its significance is slightly lower in the logistic regressions that compare inclusion on the Candidate List to inclusion on the SIN and PRIO lists, relative to the significance in the logistic regression that compares inclusion on the Candidate List to inclusion on the REACH list. A potential explanation is that the inclusion of substances on the SIN and PRIO lists is, to a large extent, based on expert judgment and scientific evidence. Nevertheless, the fact that information is statistically salient even in these samples suggests that science has an

important role to play in enabling the implementation of chemical regulation.

The relative importance of the number of countries producing or importing the substance in the EEA is somewhat surprising, particularly since its relative effect is almost three times as large as that of the CMR Score. In fact, almost half of the substances on the Candidate List −144 out of 303−are not produced or imported into the EEA. These compounds might be listed on the Candidate List because member states want to ease political opposition by proposing inclusion on the Candidate List of "low-hanging" fruit, e.g., chemicals known to be hazardous but whose listing will not affect the economic interests of other European countries.

An alternative explanation is that European countries want to prevent such substances from being produced or imported in the future, for example, as alternatives to existing SVHCs. By such listings, European countries with stricter domestic chemical regulations might avoid putting national companies at a competitive disadvantage or might ensure that firms that develop non-toxic chemical alternatives are properly rewarded. This explanation finds some support in the fact that many of the proposals for the Candidate List of chemicals not produced/imported into the EEA have been made by countries such as Germany and Sweden. These countries have consistently supported the implementation of stricter chemical regulations in Europe. This support has been partly based on a desire to export their stricter domestic chemicals standards and policies to the European level, alongside efforts to further strengthen their domestic regulations[27].

Regardless of the explanation, the evidence raises concerns about the actual extent to which the Candidate List has encouraged manufacturers to remove substances of very high concern from the market, when the production/import of almost half of the substances included on the Candidate List had either already ceased before the listing, or the compound has never been produced or imported in the European Economic Area at all.

## Disentangling the effects of carcinogenicity, mutagenicity, reproductive toxicity, and environmental properties

To explore which specific toxicological properties have the greatest effect on inclusion on the Candidate List, we run alternative logistic

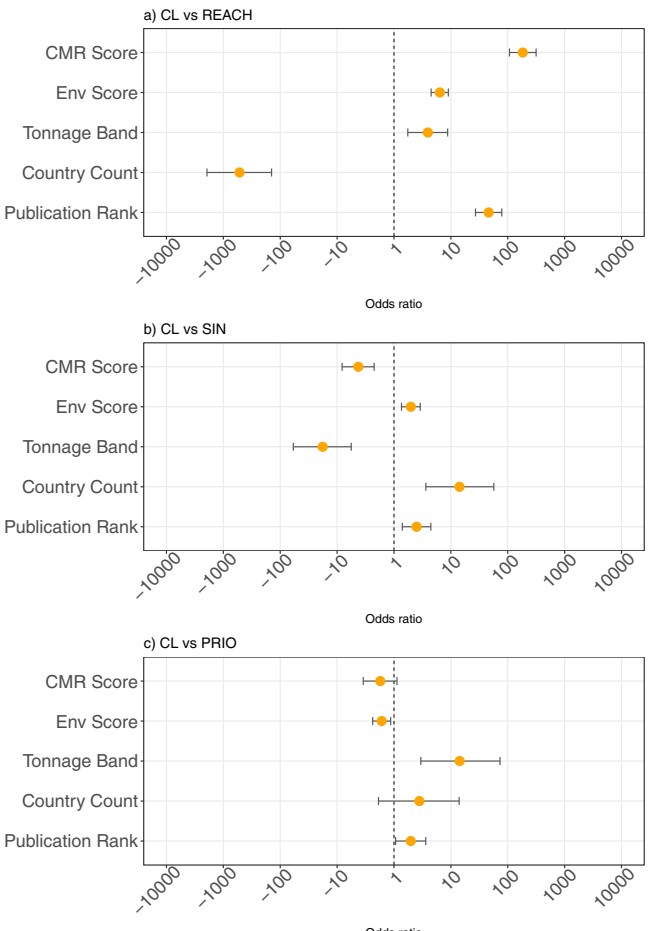

**Fig. 2 | Marginal contribution of explanatory variables to the odds of inclusion on Candidate List.** The figure presents the point estimates of the odds ratio and the corresponding 95% confidence interval for all parameters included in the logit model where the likelihood of inclusion on the Candidate List is predicted by the toxicological properties of the substances as well as the economic and knowledge parameters. Chemicals listed in the Candidate List are compared to (**a**) all chemicals registered under the European chemical regulation Registration, Evaluation, Authorization, and Restriction of Chemicals (REACH, *n* = 15,169), (**b**) chemicals listed on the SIN list (SIN, *n* = 874), and (**c**) chemicals listed on the PRIO list (PRIO, *n* = 856). Parameters with estimates to the left of the vertical line at 1 will typically be of lower value in the Candidate List as compared to the list with which it is compared, and vice versa.

regressions, where the CMR and Environmental Score are replaced by individual scores accounting for C (carcinogenicity), M (mutagenicity), and R (reproductive toxicity) properties and their associated hazard level (danger or warning), as well as individual acute and chronic environmental hazard variables. Figure 3 plots the marginal contribution and the corresponding confidence intervals of the variables in these regressions.

The results indicate that, compared to the chemical substances registered under REACH, the strongest predictor of inclusion on the Candidate List is reproductive toxicity, followed by carcinogenicity danger and chronic aquatic toxicity (Fig. 3a). Comparing the SIN list to the Candidate List also shows that carcinogenicity danger, one of the regulatory criteria for inclusion on the Candidate List, is underrepresented in the Candidate List as compared to the SIN list. A similar observation can be made when comparing the Candidate List to the PRIO list (Fig. 3b, c). In contrast, reproductive toxicity is prioritized in the Candidate List when compared to the SIN and PRIO lists.

The stronger focus of the Candidate List on chemicals that are toxic to reproduction might be explained by the growing scientific and public concern that chemicals in the environment might impair both human and wildlife reproduction[28]. This public attention has been matched by considerable political activity, especially within the European Union. For instance, reproductive toxicology has been highlighted as a prioritized area in the European Commission's White Paper on a Strategy for a Future Chemicals Policy[29].

Concerning economic motivation and information, we observe the same results as in Fig. 2. For instance, the amount of scientific information about the substance increases the odds of inclusion on the Candidate List as compared to all other lists, though the effect is particularly salient when comparing the substances listed in the Candidate List to REACH. Compared to REACH, a lack of publications on the substance reduces the odds of inclusion on the Candidate List by almost 50-fold when compared to substances that are very well studied.

Moreover, compared to REACH, the number of countries producing or using the substance in the EEA has the largest relative importance affecting the odds of inclusion on the Candidate List. For instance, if all countries in the EEA produced or used the substance, the odds of inclusion would be decreased almost 700-fold as compared to the case of no country producing or using it. In contrast, if all firms have reported reproductive danger properties, the odds increase by only about a factor of 30.

**What drives inclusion on the Authorization List?**
Our analysis so far has focused on the Candidate List rather than on the Authorization List for two reasons. First, the Candidate List is the first step in the process of inclusion on the Authorization List. As shown in Table 1, the properties of the substances on both lists are statistically similar. Second, the Candidate List includes over three times the number of substances on the Authorization List, which allows for a more robust statistical analysis of the drivers of the probability of inclusion. Nevertheless, to verify whether our results also hold for the SVHCs on the Authorization List, we perform logistic regressions to analyze the relative importance of toxicity, economics and available scientific knowledge on the probability of inclusion on the Authorization List.

The results of such regressions and accompanying figures are presented in the Supplementary Information (see Supplementary Tables 4, 5 and Supplementary Figs. 1, 2). They confirm our findings: the lack of production and import of the chemicals in the European Economic Area is also the most important determinant for inclusion on the Authorization List. Furthermore, the effects of production/imports decreasing the odds of inclusion are much more salient in the Authorization List than in the Candidate List (i.e., 50 times larger in AL than in CL), which is somewhat not surprising given that the goal of interested groups is to avoid the implementation of binding restrictions that might affect their economic interests. It seems then natural that the salience of the lack of production and import is larger for the chemicals included on the Authorization List. Indeed, out of 86 substances on the Authorization List, we find that 36 substances had zero active or inactive registrants at the time the data was collected (February 2020), despite of not having a binding sunset.

As for the relevance of C, M, R, and environmental properties, compared to the chemical substances registered under REACH, the strongest predictor of inclusion on the Authorization List is reproductive toxicity and carcinogenicity danger, followed by aquatic toxicity (Supplementary Fig. 2a). The analysis also shows that reproductive toxicity is prioritized in the Authorization List also when compared to the SIN and PRIO lists (Supplementary Fig. 2b, c). Furthermore, as in the case of the Candidate List, the number of publications have a positive effect in explaining inclusion on the Authorization List; such effect is robust across all our subsamples and

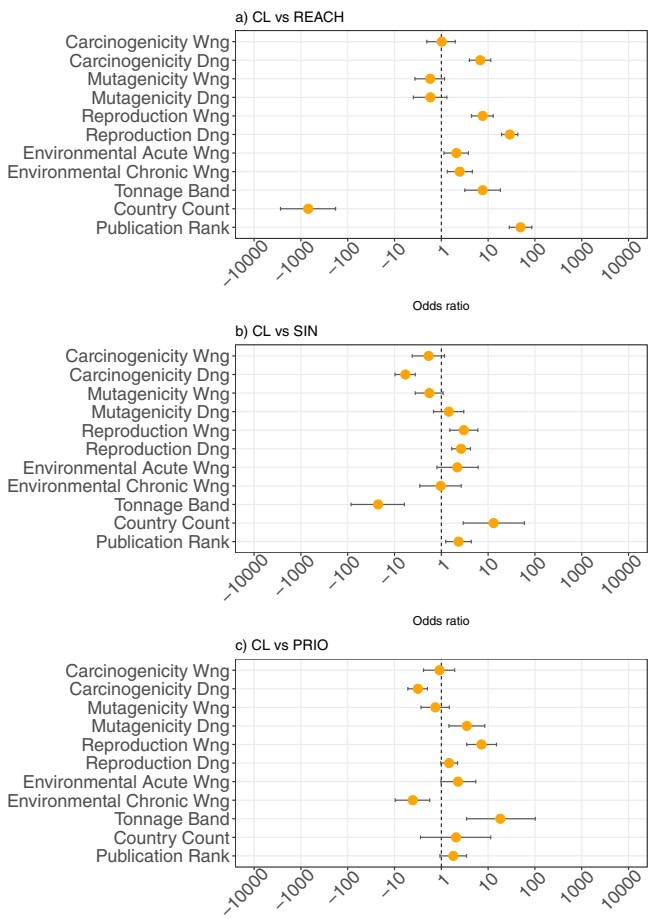

**Fig. 3 | Marginal contribution of carcinogenicity, mutagenicity, reproductive toxicity, and environmental properties to the odds of inclusion on the Candidate List.** The figure presents the point estimates of the odds ratio and the corresponding 95% confidence interval for all parameters included in the logit model where we explore in further detail which specific toxicological properties affect the listing on the Candidate List. Chemicals listed in the Candidate List are compared to (**a**) all chemicals registered under the European chemical regulation Registration, Evaluation, Authorization, and Restriction of Chemicals (REACH, *n* = 15,169), (**b**) chemicals listed on the SIN list (SIN, *n* = 874), and (**c**) chemicals listed on the PRIO list (PRIO, *n* = 856). Parameters with estimates to the left of the vertical line at 1 will typically be of lower value in the Candidate List as compared to the list with which it is compared, and vice versa.

more salient in the logistic regression that compares inclusion on the Authorization List to REACH (Supplementary Fig. 1a and Supplementary Fig. 2a).

Finally, it is worth highlighting that as anticipated, the statistical power of the analysis of the drivers of inclusion on the Authorization List is lower than for the Candidate List, which is confirmed by the larger intervals of confidence of our estimates in Supplementary Figs. 1 and 2 versus those in Figs. 2, 3.

## Discussion

REACH aims to protect public health and the environment from undue risk of harm, as well as to promote alternative test methods, the free circulation of chemical substances on the internal market and the enhancement of competitiveness and innovation. A way to protect public health and the environment is through the removal of SVHCs (substances of very high concern) from the market. Have the listing procedures that shape the Candidate List contributed to achieving these goals? In this paper, we analyze the drivers of inclusion on

REACH's Candidate List of substances of very high concern, providing clear insights on the conflict between hazard reduction and economic interests.

Naturally, we find that hazard reduction is one important driver of the inclusion on the Candidate List. The evidence presented in this paper suggests that the outcomes of the listing are consistent with the protection of human health and the environment, as chemical properties associated with both CMR (carcinogenicity, mutagenicity, and reproductive toxicity) and environmental harm are highly significant in explaining the inclusion on the Candidate List. CMR properties are a more significant predictor of inclusion on the Candidate List than environmental effects. Moreover, among CMR properties, the Candidate List has a strong focus on chemicals that are toxic to reproduction. Furthermore, the Candidate List has successfully included hazardous chemicals flagged by lists such as the SIN and PRIO lists.

Our results also confirm that science has an important role to play in enabling the implementation of chemical regulation. Unfortunately, there are many chemicals for which little or no scientific knowledge is available[30]. Thus, efforts to increase our understanding of the effects of widely used chemicals should be urgently enhanced.

However, the most important and significant variable explaining the odds of inclusion on the Candidate List is not how dangerous a chemical is but the fact that it is neither produced nor imported into the European Economic Area. The lack of production and import of the chemicals is also the most important determinant for inclusion on the Authorization List. Our model does not allow us to investigate causality, but the correlation is so strong that the question must be posed: why is this regulatory process focusing on chemicals that are neither produced nor imported in the EEA?

Indeed, even if it is not surprising that multiple factors determine the selection of chemicals, the relative importance of the number of countries producing or importing the substance in the EEA is somewhat surprising and calls into question the actual effectiveness of the Candidate List in removing SVHCs from the market. In contrast to the SIN and PRIO lists, which have no legal implications that can lead to restriction or banning of the listed substances, substances on the Candidate List may eventually be banned in Europe if they are later included on the Authorization List. This difference could explain our findings that inclusion on the Candidate List and on the Authorization List is strongly biased in favor of substances not produced or imported in the EEA.

We acknowledge that there might be other factors that can explain the listing of SVHCs that are not captured by our analysis. For instance, the details of the political process leading to the listing of the chemicals is not observable and, thus, not measurable. Nevertheless, we believe that the variables included in our analysis are sound, which is confirmed by the statistical significance of our analysis. We also acknowledge that our analysis is focused on the listing of SVHCs under the Authorization program, while hazardous substances can also be regulated under the Restriction program. The interaction between these two programs is suggested as an area for further research.

The fact that interest groups influence the design of environmental policies is well documented in the literature. It is also well documented that business and industry interests are in general in a better position to represent their interests and achieve their preferences than organizations representing diffuse interests. In the case of listing SVHCs, national interests are represented both when substances are proposed and in the deliberation process. It is not difficult to imagine the outcomes that might arise from such processes. Member states interested in the regulation of SVHCs naturally start by proposing substances for which there are well-documented effects and little political opposition due to their limited production and use within the EEA. Our results suggest that low-hanging fruit has been picked first and that it may well become increasingly difficult over time to agree on chemicals for which there is little political opposition due

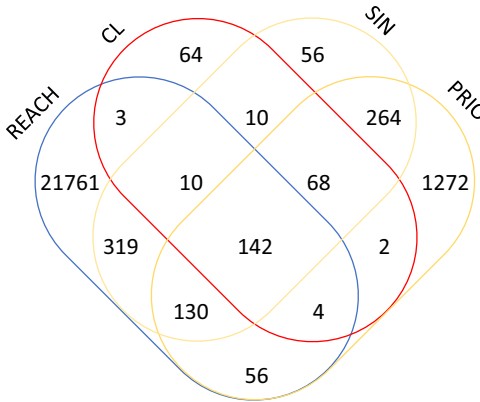

**Fig. 4 | Overlap of substances among the chemical lists.** Venn diagram representing the overlapping of chemical substances included in the European chemical regulation Registration, Evaluation, Authorization, and Restriction of Chemicals (REACH, $n = 22{,}425$), Candidate List (CL, $n = 303$), SIN List (SIN, $n = 999$), and PRIO List (PRIO, $n = 1938$).

to a lack of local production. This might require a fundamental redesign of the process to ensure that hazardousness becomes the most important driver of inclusion on the Candidate List, and that hazardous chemicals produced or imported in the EEA are listed even if they affect the economic interests of European firms.

## Methods

### Lists of Chemicals

REACH applies to chemical uses not covered by other legislation and includes chemicals used in industrial processes and intentional chemical mixtures and chemicals added to products in the European Union[6,7]. A list of all (22,425) compounds registered under REACH was collected in February 2020. In addition, information on 303 chemicals listed on the Candidate List and 86 chemicals listed on the Authorization List was collected in February 2020.

The SIN list (Substitute it Now) is an alternative list of SVHC-like compounds that fulfill the criteria for SVHC as defined in the REACH regulation. These include substances that are carcinogenic, mutagenic, or pose a risk of reproductive harm (CMR); those that are persistent, bioaccumulative, and toxic (PBT) or very persistent and very bioaccumulative (vPvB); and substances that present an equivalent level of concern. Thus, the SIN List includes chemicals that are likely to be legally restricted or require authorization under REACH. A list of all SIN compounds (999) was collected in February 2020.

The PRIO list was implemented in 2004 by the Swedish Chemical Agency (KEMI). The objective of PRIO was to increase knowledge about the handling of hazardous chemicals and to reduce their use without imposing any legal constraints. The substances listed under PRIO are prioritized under two categories: (1) phase-out substances and (2) priority risk-reduction substances. The PRIO phase-out includes substances that pose severe hazards for human health and the environment (i.e., CMR, PBT/vPvB, strongly allergenic, endocrine disrupting, fluorinated greenhouse gases, ozone depleting, particularly hazardous metals, and particularly persistent substances). Such substances may be identified as a substance of very high concern (SVHC) and may gradually be subject to permit testing in the EU chemical legislation REACH. Some of the phase-out substances are already banned or restricted in certain uses. We collected a list of all (1938) compounds which were labelled as 'phase-out' substances in PRIO in February 2020.

The overlap among all the chemicals registered under REACH and the chemicals included on the Candidate List, SIN list and PRIO list is shown in Fig. 4. Most of the substances registered under REACH

(21,761 out of 22,425) are not included on any of the lists of hazardous substances, while 142 chemical substances are included on all of them. More than 2000 substances included on the SIN or PRIO lists are not included on the Candidate List and 47 out of the 303 substances included on the Candidate List are not included on the SIN or PRIO lists. Finally, all substances on the Authorization List are also on the Candidate List.

### Data

Toxicological variables: Toxicological properties of chemicals were collected from their respective hazard classifications in accordance with the globally harmonized classification (GHS), available through the ECHA classification and labelling inventory. The chemicals in the inventory are classified by the registering firms according to their physical, health and environmental hazards by means of the applicable GHS codes. Moreover, hazards are graded from low to high using a standardized signaling word ("warning" or "danger", where "danger" is used for more severe hazard categories).

At the time the data was collected, the GHS inventory contained information for 147,633 registered compounds covering 146,597 unique European Community (EC) numbers. For each chemical, we computed the percentage of firms which label the compound with either of the GHS codes related to CMR properties or environmental hazard. We then determined the percentages of notifications individually for each specific property (property (C, M or R, for CMR Score and "Very toxic to aquatic life" or "Very toxic to aquatic life with long lasting effects" for Environmental Score). A CMR score was devised where the percentage of all firms reporting GHS codes for C, M and R properties were summed and the difference in severity between signaling words (i.e., warning and danger) was accounted for by multiplying all percentages associated with 'danger' by two. Finally, the score was normalized to vary between 0 and 1. For the environmental score, we only added the percentages related to acute or chronic hazard without weighting, because all relevant environmentally related GHS codes used the same signaling word. The environmental score was also normalized to a score between 0 and 1.

It should be noted here that the underlying GHS classifications for environmental hazard account for persistence (P) and bioaccumulation (B). The classification depends on the toxicity of the compound, where a compound is labelled as hazardous to the environment at different levels of toxicity depending on its P and B properties. For instance, a rapidly degradable compound which does not bioaccumulate will be classified as GHS410 (Long-Term Aquatic Hazard; signal word: warning) if effects can be seen at concentration below 0.01 mg/L for algae, crustaceans, or fish. If the compound is not rapidly degradable, the classification applies if effects are seen at concentrations below 0.1 mg/L, and if the compound is also bioaccumulative the classification applies if effects are seen at concentrations below 1 mg/L[31]. Thus, less toxicity is required for the classification if the compound is also P and B.

Economic variables: Information on the production/import tonnage and number of countries with production/import within the EEA was collected from the REACH Registration dossiers. A country is reported as having production/import when there are active or inactive registrants for the substance. An inactive registrant has produced/imported the substance in the past but ceased production/ import at the time of REACH registration. Unfortunately, we cannot disentangle production from imports. It is possible that opposition might be stronger when a country produces the chemical. On the other hand, because REACH addresses industrial chemicals, imports of chemicals might affect the interests of the countries, as national firms might be using chemicals in their production processes. If so, inclusion on the Candidate List – and eventually on the Authorization List – could affect their access to a relevant production input.

The number of countries was square root-transformed to reduce the skewness of the data. The tonnage band of each compound was log10 transformed using the upper reported tonnage band (e.g., a compound reported as having a tonnage band 10–100 was transformed to 2). If a substance had multiple entries in the REACH registry, only the largest tonnage was used. Compounds with no production or only intermediate use were treated as having zero value, while substances with only confidential tonnages were considered to have no data available.

Knowledge variables: We collected information on the number of publications available per chemical in journals related to environmental and human toxicology via the online database PubMed. Information could be collected for a total of 18,809 of the 25,875 compounds included in this study. For each compound, ten of the most used chemical names were extracted from PubChem[32], which resulted in 117,486 names in total. These names were searched for in all publications in the last 20 years (2000–2019) in 36 selected international peer-reviewed toxicological and ecotoxicological journals available in PubMed. The journals were chosen to (1) provide representation in the fields of toxicology and ecotoxicology, (2) publish mainly original research, (3) have an international focus and a peer-review principle, and (4) have overall high scientific standards (thus excluding any potentially predatory journals). These criteria were manually assessed based on information from the journals' homepages and bibliometric information from the Scopus and Web of Science databases. This resulted in 194,018 papers. We searched the title, abstract, keywords and chemical lists of these papers for the occurrence of any of the compound names. In total, 111,284 papers were found to contain 248,907 occurrences of a total of 4,088 non-redundant compounds.

The total knowledge for each compound was quantified by counting the number of publications in which it appeared. Since the distribution of the number of publications is very skewed, we made use of a publication rank based on a grouped rank that depends on the number of publications available. The group rank variable varies between zero and four, where zero denotes a compound with no information, and a score of 1, 2, 3, and 4 is given for compounds occurring in 1–10, 11–100, 101–1000 and more than 1000 publications respectively.

Supplementary Table 1 summarizes all the information collected. A list of the 36 selected international peer-reviewed toxicological and ecotoxicological journals utilized to construct our proxy for scientific knowledge available is also available in the Supplementary Information as Supplementary Method 1.

### Logistic regression and variable significance

We investigate the drivers of inclusion on the Candidate List and on the Authorization List by means of logistic regressions, where the outcome variable (i.e., inclusion on the Candidate List or inclusion on the Authorization List) is a binary variable explained by a linear combination of a series of predictors, i.e., toxicological, economic and availability of scientific knowledge. Logistic models were fitted independently to the substances on the Candidate List versus all compounds included in REACH, the SIN list, and the PRIO list. The same procedure was followed for the substances on the Authorization List. The analysis was performed using the R v3.5.1 statistical programming language. The analysis and visualization was aided using the public libraries ggplot2 v3.3.5, tidyverse v1.3.1 and stringr v1.4.0.

Initial assessments were performed to remove variables with high correlation because such variables make it difficult to interpret the modelling. The fit of the models was determined using the adjusted McFadden pseudo $R^2$, which penalizes model overfitting by accounting for the number of variables included in the modelling.

Tests for variable significance were performed using the variables' independent fit for a logistic model against the null model.

Supplementary Tables 2, 3 in the Supplementary Information present the results of the logistic regressions for the Candidate List, while Supplementary Tables 4, 5 present the results of the logistic regressions for the Authorization List.

### Reporting summary

Further information on research design is available in the Nature Portfolio Reporting Summary linked to this article.

## Data availability

Data generated in this study have been deposited in the Zenodo data repository; Data for "Economic Interests Cloud Hazard Reductions in the European Regulation of Substances of Very High Concern"; https://doi.org/10.5281/zenodo.7051114.

## Code availability

The codes utilized to analyze the data that supports the findings of this study are available in the Zenodo data repository; Data for "Economic Interests Cloud Hazard Reductions in the European Regulation of Substances of Very High Concern"; https://doi.org/10.5281/zenodo.7051114.

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

## Acknowledgements

Funding from the FRAM Centre for Future Chemical Risk Assessment and Management at the University of Gothenburg was received by J.C., E.K., and M.G. Funding from the Swedish Research Council was received by J.C., E.K., and M.G.

## Author contributions

J.C.: Conceptualization, Methodology, Formal analysis, Validation, Writing—Original Draft and Review & Editing, Visualization, Funding acquisition. E.K.: Conceptualization, Methodology, Validation, Writing—Review & Editing, Funding acquisition. M.G.: Conceptualization, Methodology, Data Curation, Validation, Visualization, Writing—Review & Editing.

## Funding

## Competing interests
The authors declare no competing interests.
