## [Peer Review File · Nature Communications]

Economic Interests Cloud Hazard Reductions in the European Regulation of Substances of Very High ConcernReviewers' Comments:

Reviewer #1:

Remarks to the Author:

This is an interesting paper that will make a valuable contribution to the literature. The authors ask a seemingly obvious, but to my knowledge largely unexplored question – namely, is the EU's Candidate List of SVHC's Chemicals different from other lists of highly toxic chemicals compiled by other expert bodies. Their methods, which compare a range of variables and develop novel ways to quantify scientific information (like warnings), seem robust. And their findings are impactful and will undoubtedly interest a wide range of readers who will be eager to learn still more about the EU prioritization process governing SVHC chemicals.

I do not have the statistical background to comment on the mathematical features of the paper. Instead, my comments identify a few spots that deserve a bit more discussion, even if it amounts to little more than explaining why the authors chose not to elaborate or investigate the issue further.

- The authors note that one source suggests there should be 1500 SVHC listed chemicals (I'm not sure if this is the Candidate or Authorization List?). Instead, there are far fewer chemicals on the Candidate List and only 86 on the Authorization List. Since the title and introductory material of the article is focused on regulation of SVHC chemicals, the small numbers of listed chemicals may deserve a bit more discussion to broaden the perspective beyond simply examining the characteristics of listed chemicals. Is it helpful to compare the number of CL chemicals against the number of chemicals on the SIN list for example? Or some other benchmark? The small number on the CL and AL lists obviously reinforces the findings of the paper, but from a different angle.
- There are comparisons between various chemical lists and the CL, but only Table 1 offers any comparisons between the CL and AL and there is no discussion of these results. I assume that is because there was no statistical significance. But perhaps there would be with comparisons between the AL and other lists? Whatever the case, it would be helpful to let the readers know why the authors focus primarily on the CL rather than AL, even though the AL is the list that leads to binding requirements. Given the more elaborate consultation processes governing the AL list, coupled with the binding requirements, it is also the set of chemicals for which we would expect the most industry influence.
- Readers may want to better understand how the authors selected the comparison lists like SIN (perhaps these are the only lists that can serve as benchmarks)? A little more discussion on why these lists and not others (if there are any) were chosen would be helpful.
- The criteria for the Candidate List (provide a dossier, host a consultation, and list if there is no objection or a committee recommends listing) fit hand in glove with the empirical findings of a publication-bias and the limited EU use of the listed chemicals. It may be worth making those links between decision criteria and outcomes even clearer in the final discussion section.
- This comment goes beyond the paper, but it would have been fascinating to see the chemicals that are nominated for the CL (before consultation) and compare them to those that make it on the CL. Are chemicals that are in wider use getting nominated, but drop out in part because of the criteria for listing (e.g., no objection)? Perhaps 1000s of chemicals are being nominated for the CL, for example, and the majority fall out because of the consultation and "no objection" criteria. Or conversely, perhaps interest groups and economic concerns affect the nomination process itself, and most of the nominated chemicals ultimately do make it through to the Candidate List. This added information would also inform the kind of reforms needed to the EU's listing process. Even providing the raw percentages of chemicals on the CL versus the number that were nominated would be helpful, without more detailed analysis.

Thank you for a fascinating paper!

Reviewer #2:

Remarks to the Author:

Given the importance of chemicals in our society and of REACH as the major instrument for regulating the health and environment risks associated with them in the EU, this paper raises a highly relevant question. Moreover, the outcomes – particularly those related to the significance of economic parameters in including substances on the Candidate List – are illuminating and a reason for serious concern about the effectiveness of the REACH procedures in terms of health and environment.

Before turning to my comments, I have to state that I am not an economist nor a quantitative (social science) researcher. For this reason I will not be able to critically assess the 'technicalities' of the data analysis. I trust other reviewers will take care of this (important!) aspect of the paper.

This having said, I do have a number of comments and suggestions which may help to improve the paper:

1. The Intro section provides a good and helpful overview of REACH and the relevant procedures as well as a brief review of mainly political science literature regarding REACH and chemicals regulation more generally. This explains and justifies the research puzzle in a general sense. What I miss, however, is a more focused underpinning of the selection of the key variables and hypotheses underlying this research (lines 133-38). Why have the researchers decided to focus on toxicological properties, economic motivations and scientific knowledge in the first place? Why not, for instance, on the type, or the type of use, of the chemicals involved? One may well imagine that agricultural chemicals lead to a different 'dynamic' under REACH than for instance food additives or chemicals used for the production of pharmaceuticals. And perhaps even more importantly, why have the three key variables (toxicological properties, economic motivations, and available scientific knowledge) been operationalised in the way described (but not explained) in the 'Materials and methods' section and what are the assumptions underlying them? E.g.: why do you suspect that 'chemicals produced or used in large quantities or by many EU countries will be more difficult to regulate' (line 136-7)? What is the argument? Is this underpinned by literature? A more thorough discussion of these points in the first part of the paper would also provide a fertile basis for a more profound and literature-based discussion at the end (see also point 3, below).
2. More specifically, regarding the economic variable, I wonder about the distinction between 'production' and 'use' of chemicals inside/outside the EEA. 'Production' and 'use' are now sometimes lumped together and sometimes used interchangeably in the text. There are good reasons to assume that the interests of countries producing a chemical substance are in fact quite different from those of countries just using it. It would be helpful, and probably provide additional insight, if this distinction was analysed more explicitly. At present, it is not even clear to me what the data used now ('production/use tonnage and number of countries with production/use', lines 451-9) exactly refer to. In any case, this could be better explained.
3. Currently, there is a certain disagreement between the 'Significance statement' and the Intro (particularly lines 113-23), which put considerable emphasis on the influence of business interests and advocacy, and the actual analysis, which does not go into these factors at all. I can of course accept that an analysis like the present one 'does not allow (...) to investigate causality' (lines 351-2), but now the 'Significance statement' and the Intro seem to promise more than the paper actually delivers. This could be repaired either by rephrasing the 'Significance statement' and the Intro, or by making a more serious attempt in the 'Discussion and Conclusion' to link back the findings of this research to the literature on interest representation and advocacy cited in the Intro. (I would prefer the latter solution...)
4. Lines 269-73 suggest an interesting alternative explanation for the finding that a large part of the listed chemicals is produced outside the EEA. I think this explanation deserves more attention, as it may well play a role for at least part of the chemicals involved. I am not an expert in the chemicals field, but I am quite sure that there must be some literature analysing the effect of more stringent policies in leader countries (including Sweden) on the implementation of REACH. It would give a touch of (environmental) optimism to the (basically justified) concern about this particular finding.
5. Lines 277-78 state that 'the production of almost half of the substances included on the Candidate

List 278 had already ceased before the listing' (also in the abstract, line 37). As far as I can see, this can not be derived from the data. Isn't it just as well possible that the chemicals involved had never been produced in the EEA at all? If this were the case for a reasonable number of chemicals, this would further nuance the picture sketched in the paper (see previous point).

6. Finally, for a non-specialist reader (and as far as word count allows), some parts of the results section would benefit from more explicit explanation of what we actually see in the figures and tables (e.g Fig. 2, lines 207ff).

All in all, a highly relevant, thought-provoking and generally well-written paper. However, and stressing once again that the technical aspects of the data analysis are outside my scope of expertise, I see room for improvement in the argumentation, as detailed above. Therefore, my recommendation is: revise and resubmit.

Reviewer #3:

Remarks to the Author:

The paper offers a quantitative analysis of the determinants for placing most hazardous chemicals on the REACH Candidate list for Authorisation. This is an interesting and relevant topic considering in particular the ambitious goals of the European Union's Green Deal, i.e. to reach a pollution-free environment, and the corresponding review of several European regulatory frameworks, including REACH.

The authors apply logistic regression to different lists of hazardous chemicals in order to examine the relative importance of selected determinants for inclusion on a list. This offers interesting and novel insights in the drivers of chemicals' selection for being shortlisted for regulatory action under REACH. The authors conclude that the relevance of economic determinants reflects a systematic bias towards chemicals which are of minor relevance in the EU, and, in turn, for regulation.

While I consider this an interesting study, I think the authors' conclusions are too far-fetched, and may even misinterpret the functioning of the REACH regulatory framework.

Below I explain my arguments in more detail, providing also suggestions for further improving the manuscript.

1) Focus of the study:

1.1) The authors focus on authorisation as one of the two regulatory instruments of REACH (with restrictions being the second instrument). This is reasonable for the analysis but also bears a risk of ignoring that authorisation and restriction are not separate regulatory steps, but inter-connected and complementary. In fact, while a restriction procedure can be initiated for any chemical (not just SVHC), member states can decide to start a restriction procedure on a chemical included on the candidate list if it appears that an authorisation procedure may not sufficiently cover the risk. For instance, this is the case if a SVHC is not widely produced in the EU, but imported in large quantities from outside the EU. A recent and prominent example is Dechlorane Plus, which was included on the Candidate list in 2018 due to its vPvB and potentially ED properties (see ECHA Candidate list table). This chemical is only produced outside the EU, still, it is highly economically relevant in the automotive and aviation industry. The Norwegian competent authorities, therefore, started a restriction procedure in 2020. The authors may refer to the restriction proposal on the ECHA website, and a corresponding Eftec report prepared under the Stockholm Convention, for further information. In summary, it is not correct that inclusion of a chemical on the Candidate list is a one-way road to authorisation. As a consequence, it is not justified to conclude that chemicals that enter the Candidate list which are not produced in the EU are necessarily 'of secondary importance'.

1.2) Note, further, that inclusion of a SVHC on the candidate list is just the first step for triggering authorisation. From the Candidate list, chemicals will be prioritised to be included on the Authorisation list'. Prioritisation occurs via a criteria-based selection process, using EU tonnage volume as an approximative criterion for both economic relevance and exposure (see

https://echa.europa.eu/documents/10162/17232/recom_gen_approach_svhc_prior_2020_en.pdf/fbbd748b-22dc-38c2-9b4c-58c6bc80c930).

I therefore advise the authors to either better embed their results in the regulatory context of REACH, or consider to conduct their analysis for SVHC placed on the authorisation list only.

2) Selection of toxicological variables: The current analysis uses variables referring to CMR properties and environmental properties, in particular acute and chronic aquatic toxicity. While the human health related hazard categories match those also used for the selection of SVHC under REACH, the environmental indicators used in the study don't. The identification of a SVHC is based on hazard categories explained in REACH Article 57a-e. The corresponding identification criteria are listed in the Guidance on SVHC identification (see

https://echa.europa.eu/documents/10162/2324906/svhc_en.pdf/8faef33c-b46e-4186-8b7c-8cfbeccd0812). Specifically, the combination of persistence, bioaccumulation and/or toxicity (PBT), or the combination of high persistence and toxicity is crucial for SVHC classification because it points to long-lasting and potentially irreversible impacts to ecosystems. In the current analysis, however, environmental hazards are reflected by toxicity indicators ONLY. Therefore, and given the focus on the REACH Candidate list in comparison to other existing lists of harmful chemicals, the analysis is of little informational value. I recommend to re-do the analysis and explicitly consider the combination of P, B and/or T indicators to better capture the environmental concerns in the regression analysis. It may also be necessary to re-calculate the environmental score.

3) Scoring of human and environmental hazard variables: The authors explain in line 431 onwards that human health and environmental hazard indicators were transformed into scores. I fail to understand the method for determining the score. The authors should provide a more transparent description of their approach, maybe in the Supplementary Materials section of the paper. Furthermore, I miss a list of score values for the chemicals included in the analysis under each list.

4) Comparison of results across lists: It is interesting to compare the determinants across different lists of SVHC in Europe. But the inclusion criteria for SVHC may differ between the Candidate list, SIN and PRIO. The authors should explain these criteria.

5) Documentation of data: Maybe I overlooked it but I failed to find a documentation of all data used for the analysis (in particular the list of chemicals used, and the corresponding values of the variables). To make the analysis traceable and transparent please add this information (either as part of the supplementary material, or as a separate online data repository).

6) Additional suggestions:

-Line 2/3: Given the points explained above I would suggest to tone down and re-formulate the title of the paper. The results cannot be interpreted as 'determinants of the regulation of SVHC in Europe' considering that the Candidate list is just one step in the selection process and authorisation is one of two complementary instruments for regulatory control.

-Lines 77/82: It is too vague to describe PBT/vPvB properties as posing 'broad environmental concerns'. To the contrary, PBT/vPvB denote hazard categories indicating composite hazards and, thus damage to ecosystems over time. The formulation should become more specific.

-Lines 261-278 and lines 348-353: I suggest to tone down these text passages. Looking at the result, one may conclude that 'absence from production in Europe' is the 'most significant variable' determining inclusion on the Candidate list. Considering that chemicals listed under the Candidate list can also be selected for restriction, the conclusion that the Candidate list is systematically biased towards chemicals which are economically not relevant in the EU is not warranted. See also the arguments provided under 1).

-Line 385/386: Though restriction is not limited to SVHC, past years' restriction procedures have often focused on SVHCs. Moreover, several restrictions were initiated following-up on authorisation procedures, e.g. because the Commission concluded that there is a risk of a continued use of the chemical for the EU as a whole, or because a refused authorisation does not prevent imports of a

SVHC into the EU.

-Line 467: Please explain how scientific journals were selected.

REVIEWER COMMENTS

Reviewer #1

1) The authors note that one source suggests there should be 1500 SVHC listed chemicals (I'm not sure if this is the Candidate or Authorization List?). Instead, there are far fewer chemicals on the Candidate List and only 86 on the Authorization List. Since the title and introductory material of the article is focused on regulation of SVHC chemicals, the small numbers of listed chemicals may deserve a bit more discussion to broaden the perspective beyond simply examining the characteristics of listed chemicals. Is it helpful to compare the number of CL chemicals against the number of chemicals on the SIN list for example? Or some other benchmark? The small number on the CL and AL lists obviously reinforces the findings of the paper, but from a different angle.

Rp: Estimates on the number of SVHCs come from a report from the European Commission (2013). See:

EC (European Commission). 2013. Roadmap on Substances of Very High Concern. Council of the European Union, Brussels, 6 February, 2013.

https://echa.europa.eu/documents/10162/19126370/svhc_roadmap_implementation_plan_en.pdf

We have clarified this in the text. Furthermore, as stated in the text, as of February 2020, 303 substances were listed as substance of very high concern (SVHC) on the Candidate List, and 86 of those were on the Authorization List. The low number of SVHC substances listed so far raises concerns about whether the listing procedures can ensure that the risks posed by SVHC are adequately and timely controlled, and it is a motivation for this study.

Alternative lists of hazardous chemicals in use in Europe, such as the SIN (Substitute it Now) and PRIO lists, include a much larger number of substances than the Candidate List. For instance, as of February 2020, 999 and 1938 substances were listed by the SIN and PRIO lists, respectively. We chose the SIN List and PRIO List as benchmarks for the listing of substances of very high concern in Europe because such lists follow the same technical criteria specified by REACH as a basis for the listing of substances.

As explained in the section Material and Methods, the SIN List is an alternative, legally non-binding list of hazardous chemical substances developed by the International Chemical Secretariat (Chemsec). The SIN List is compiled according to REACH criteria, and thus lists chemicals that are likely to become subject to REACH requirements.

The PRIO Phase-out List was developed by the Swedish Chemical Agency. As explicitly stated by the Swedish Chemical Agency, phase-out substances pose severe hazardous properties for human health and the environment (i.e., CMR, PBT/vPvB, strongly allergenic, endocrine disrupting properties, fluorinated greenhouse gases, ozone depleting substances, particularly hazardous metals, and particularly persistent substances). Such substances may be identified as SVHC and may eventually be subject to permit testing under the EU chemicals legislation REACH. Some of the phase-out substances are already banned or restricted in certain uses. Thus, this provides us with a relevant counterfactual of highly hazardous substances.

Why do the SIN and PRIO list include a larger number of substances than the Candidate and Authorization List? As explicitly stated by Chemsec, the SIN List contains substances that should be

listed by REACH as substances of very high concern *in the absence of political roadblocks*. The same applies to the PRIO list.

Concerning other lists, an alternative benchmark could have been to compare the chemicals listed by the Candidate List to those listed under the Toxic Substance Control Act (TSCA) in the United States. However, this list is much less relevant in the context of our study because we investigate the drivers of European regulation, and thus prefer to use lists of hazardous substances in use in Europe. The comparison of TSCA and REACH would be relevant in the context of an international comparison, but this not within the scope of the current study.

Please see also our response to your third comment.

2) There are comparisons between various chemical lists and the CL, but only Table 1 offers any comparisons between the CL and AL and there is no discussion of these results. I assume that is because there was no statistical significance. But perhaps there would be with comparisons between the AL and other lists? Whatever the case, it would be helpful to let the readers know why the authors focus primarily on the CL rather than AL, even though the AL is the list that leads to binding requirements. Given the more elaborate consultation processes governing the AL list, coupled with the binding requirements, it is also the set of chemicals for which we would expect the most industry influence.

Rp: As the reviewer suggests, we do not discuss the differences between the results for CL and AL because there are no statistical differences between the chemicals listed in the CL and AL. We now explain this explicitly in the text.

We focus on the comparison of CL (rather than AL) to the other lists, for two reasons. First, the Candidate List is a first step in the process of inclusion on the Authorization List. We expect the properties of the substances on both lists to be similar, because the Authorization List includes a subset of the chemicals included on the Candidate List. Second, the Candidate List includes over three times the number of substances on the Authorization List, which allows for a more robust statistical analysis of the drivers of the regulation.

We describe our reasoning in more detail in the revised version of the paper:

“We focus on the analysis of the Candidate List rather than the Authorization List for two reasons. First, the Candidate List is the first step in the process of inclusion on the Authorization List. As shown in Table 1, the properties of the substances on both lists are statistically similar. Second, the Candidate List includes over three times more substances than the Authorization List, which allows for a more robust statistical analysis of the drivers of the probability of inclusion.”

As mentioned in the response to your previous comment, concerning other lists, an alternative benchmark could have been to compare the chemicals listed by the Candidate List to those listed under the Toxic Substance Control Act (TSCA) in the United States. However, this list is less relevant in the context of our study since we investigate the drivers of European regulation, and thus prefer to use lists of hazardous substances in use in Europe.

3) Readers may want to better understand how the authors selected the comparison lists like SIN (perhaps these are the only lists that can serve as benchmarks)? A little more discussion on why these lists and not others (if there are any) were chosen would be helpful.

Rp. As mentioned in our response to the first comment, the reason to choose the SIN List and PRIO List is that they are explicitly designed to cover chemicals that qualify as substances of very high concern according to the criteria specified by the REACH regulation. However, in contrast to the Candidate List, which is the result of a participatory process in which different stakeholders can shape the outcome of the list, the SIN and PRIO List are elaborated by experts, and, hence, less prone to political roadblocks.

We have now added in the text:

“Both the SIN and PRIO lists have the explicit aim of raising awareness about chemicals that qualify as substances of very high concern according to the criteria specified by the REACH regulation. However, in contrast to the Candidate List, which is the result of a participatory process in which different stakeholders can shape the outcome of the list, the SIN and PRIO List are elaborated by experts, and, hence, less prone to political roadblocks.”

4) The criteria for the Candidate List (provide a dossier, host a consultation, and list if there is no objection or a committee recommends listing) fit hand in glove with the empirical findings of a publication-bias and the limited EU use of the listed chemicals. It may be worth making those links between decision criteria and outcomes even clearer in the final discussion section.

We agree with the reviewer and in the revised version of the paper, one of the paragraphs in the discussion section has been revised to account for this comment:

“The fact that interest groups influence the design of environmental policies is well documented in the literature. In the case of the listing of SVHCs, national interests are represented both when proposing substances and in the deliberation process. It is not difficult to imagine the outcomes that might arise from such processes. Member states interested in the regulation of SVHC naturally start by proposing substances for which there are well-documented effects and little political opposition due to their limited production and use within the EU. Our results suggest that low-hanging fruit has been picked first and that it may well become increasingly difficult over time to agree on chemicals for which there is little political opposition due to a lack of local production. This will require a fundamental redesign of this process to ensure that toxicity becomes the most important driver of inclusion on the Candidate List, and that hazardous chemicals produced or imported in the EU are listed even if they affect the economic interests of European firms.”

5) This comment goes beyond the paper, but it would have been fascinating to see the chemicals that are nominated for the CL (before consultation) and compare them to those that make it on the CL. Are chemicals that are in wider use getting nominated, but drop out in part because of the criteria for listing (e.g., no objection)? Perhaps 1000s of chemicals are being nominated for the CL, for example, and the majority fall out because of the consultation and “no objection” criteria. Or conversely, perhaps interest groups and economic concerns affect the nomination process itself, and most of the nominated chemicals ultimately do make it through to the Candidate List. This added information would also inform the kind of reforms needed to the EU’s listing process. Even providing the raw percentages of chemicals on the CL versus the number that were nominated would be helpful, without more detailed analysis.

Rp: To date, most of the chemicals that have been suggested for the Candidate List end up being listed. The few exceptions concern dossiers that have been withdrawn from the process and chemicals not identified as SVHCs (i.e., about 6% of the dossiers submitted). The committee of national representatives has agreed on identification in most of the cases, with only a few cases in which the chemicals have been identified as SVHCs despite the objections of a minority (about 4% of the dossiers). Thus, most of the chemicals suggested are being listed, possibly because they are not expected to have relatively large negative effects on stakeholders. However, as the reviewer suggests in a previous comment, there is an interesting political economy process from the listing on the Candidate List, to the prioritization of substances for inclusion on the Authorization List, to the final inclusion on the Authorization List. This political economy process is beyond the scope of this paper and is left as an area for further research.

In the revised version of the paper, the text has been revised to:

“As of February 2020, 303 substances were listed as SVHCs on the Candidate List, and 86 of those were on the Authorization List. In contrast, in 2013 the European Chemical Agency predicted that there should be 1,500 SVHCs addressed by REACH [11]. To date, most of the chemicals that have been suggested to the Candidate List have been listed (i.e., only about 6% of the dossiers submitted have been withdrawn or the substances not identified as SVHCs). Moreover, the committee of national representatives has mostly unanimously agreed on identification of SVHCs (i.e., only about 4% of the dossiers submitted were resolved without full agreement). However, the low number of SVHC substances listed so far raises concerns about whether the listing procedures can adequately and timely control the risks posed by SVHC. It also raises the question of whether the Candidate List is being shaped by those interest groups that are more successful in translating their preferences into policy outcomes [12, 13].

6) Thank you for a fascinating paper!

Rp: Many thanks for your constructive comments that have helped us further improve the manuscript.

Reviewer #2

1. I miss a more focused underpinning of the selection of the key variables and hypotheses underlying this research (lines 133-38). Why have the researchers decided to focus on toxicological properties, economic motivations and scientific knowledge in the first place? Why not, for instance, on the type, or the type of use, of the chemicals involved? One may well imagine that agricultural chemicals lead to a different 'dynamic' under REACH than for instance food additives or chemicals used for the production of pharmaceuticals. And perhaps even more importantly, why have the three key variables (toxicological properties, economic motivations, and available scientific knowledge) been operationalized in the way described (but not explained) in the 'Materials and methods' section and what are the assumptions underlying them? E.g.: why do you suspect that 'chemicals produced or used in large quantities or by many EU countries will be more difficult to regulate' (line 136-7)? What is the argument? Is this underpinned by literature? A more thorough discussion of these points in the first part of the paper would also provide a fertile basis for a more profound and literature-based discussion at the end (see also point 3, below).

Rp: Note that REACH addresses chemicals used in industrial processes, as well as intentional chemical mixtures and chemicals added to products in the European Union. Thus, REACH does not address substances covered by more specific regulations (such as medicines and agricultural chemicals). We have re-written the corresponding paragraph in the introduction to make this clearer. We agree that the other types of chemicals are very relevant. However, this is left for further research, because the principles that govern the regulation of other types of chemicals, such as pesticides and cosmetics, are different from those used by REACH.

Concerning our choice of variables, this is based on the explicit purpose of REACH, which is to reduce hazards to both public health and the environment, and on a review of relevant literature. Since substances of very high concern are defined based on their toxicological properties, we expect such properties to be a clear determinant of inclusion on the Candidate List.

Concerning economic variables, the fact that interest groups influence the final design of environmental policies is well documented. Fuel taxation is a good example of how politically difficult it can be to implement policies that affect the economic interests of many stakeholders, even when such policies are environmentally beneficial. For instance, Hammar et al. (2004) investigate the political economy of fuel taxation and find that high levels of fuel consumption make consumers more likely to oppose increased fuel taxes, whereas consumers are more likely to tolerate them at lower levels of fuel consumption. Climate policy provides another good example. Kolk and Pinkse (2007) analyze multinational corporations' political activities on climate change and show that many firms push policy makers in the direction of policies that would allow some segments of polluting industries to gain rather than lose. Such evidence is consistent with our findings. We operationalize economic interest by means of tonnage band and by the number of countries in the European Economic Area with REACH registrants. These are good proxies for the number of stakeholders (i.e., number of countries) affected by the listing, as well as the extent to which they are affected (i.e., tonnage band). These are also the best economic variables available through REACH.

Concerning knowledge, it is well documented that science provides decisionmakers with a source of legitimation for their actions, and that regulatory actions are often focused on controlling high-profile chemicals risks. Thus, we anticipate that widely studied chemicals are easier to regulate than less studied chemicals.

We have re-written the text that specifies the aim of the study. It now reads:

“In this paper, we investigate the relative importance of toxicological properties, economic motivations, and available scientific knowledge as drivers of the listing of SVHCs on the Candidate List. Our choice of variables is based on the stated goal of REACH, which is to reduce hazards to both public health and the environment, and on a review of relevant literature. First, since SVHCs are explicitly defined based on their toxicological properties, we investigate whether the toxicity of a chemical substance is the main determinant of inclusion on the Candidate List. Second, we suspect that chemicals produced or used in large quantities or by many EU countries will be more difficult to regulate. We believe this is the case because of extensive documentation that it is politically difficult to implement policies that affect the economic interests of many stakeholders, and that industrial stakeholders try to shape policies to reduce harm to the industry (see e.g., [24, 25]), Third, since it well known that scientific evidence provides decisionmakers with political support for policy implementation [26, 27], we suspect that widely studied chemicals are easier to regulate. “

Moreover, in the discussion, we acknowledge that there might be other factors that can explain inclusion on the Candidate List that are not captured by our analysis. For instance, the details of the political process leading to the listing of the chemicals is not observable and, therefore, not measurable. Nevertheless, we believe that the variables included are sound, which is reflected by the statistical significances that we observe in our analysis.

2. Regarding the economic variable, I wonder about the distinction between ‘production’ and ‘use’ of chemicals inside/outside the EEA. ‘Production’ and ‘use’ are now sometimes lumped together and sometimes used interchangeably in the text. There are good reasons to assume that the interests of countries producing a chemical substance are in fact quite different from those of countries just using it. It would be helpful, and probably provide additional insight, if this distinction was analysed more explicitly. At present, it is not even clear to me what the data used now (‘production/use tonnage and number of countries with production/use’, lines 451-9) exactly refer to. In any case, this could be better explained.

Rp. Information on the production/import, tonnage, and number of countries with production/import within the EEA was collected from the REACH Registration dossiers. A country is reported as having production/import when there are active or inactive registrants for the substance. An inactive registrant has produced/imported the substance in the past but ceased production/ import at the time of REACH registration. Unfortunately, we cannot disentangle production from imports. As the reviewer suggests, opposition might be stronger when a country produces the chemical. On the other hand, because REACH addresses industrial chemicals, imports of chemicals also might affect the interests of the countries because national firms might be using chemicals in their production processes. In that case, inclusion on the Candidate List – and eventually on the Authorization List – would affect their access to a relevant production input.

In the revised version, we have clarified that our analysis concerns production and imports.

3. Currently, there is a certain disagreement between the ‘Significance statement’ and the Intro (particularly lines 113-23), which put considerable emphasis on the influence of business interests and advocacy, and the actual analysis, which does not go into these factors at all. I can of course accept that an analysis like the present one ‘does not allow (...) to investigate causality’ (lines 351-2), but now the ‘Significance statement’ and the Intro seem to promise more than the paper actually delivers. This could be repaired either by rephrasing the ‘Significance statement’ and the Intro, or by making a more

serious attempt in the 'Discussion and Conclusion' to link back the findings of this research to the literature on interest representation and advocacy cited in the Intro. (I would prefer the latter solution...)

Rp: In the revised version of the paper, we do so. The final paragraph of the paper reads now as:

"The fact that interest groups influence the design of environmental policies is well documented in the literature. In the case of the listing of SVHCs, national interests are represented both when proposing substances and in the deliberation process. It is not difficult to imagine the outcomes that might arise from such processes. Member states interested in the regulation of SVHC naturally start by proposing substances for which there are well-documented effects and little political opposition due to their limited production and use within the EU. Our results suggest that low-hanging fruit has been picked first and that it may well become increasingly difficult over time to agree on chemicals for which there is little political opposition due to a lack of local production. This will require a fundamental redesign of this process to ensure that toxicity becomes the most important driver of inclusion on the Candidate List, and that hazardous chemicals produced or imported in the EU are listed even if they affect the economic interests of European firms."

4. Lines 269-73 suggest an interesting alternative explanation for the finding that a large part of the listed chemicals is produced outside the EEA. I think this explanation deserves more attention, as it may well play a role for at least part of the chemicals involved. I am not an expert in the chemicals field, but I am quite sure that there must be some literature analyzing the effect of more stringent policies in leader countries (including Sweden) on the implementation of REACH. It would give a touch of (environmental) optimism to the (basically justified) concern about this finding.

Rp: As stated in the revised text, this explanation finds some support in the fact that many of the proposals for the Candidate List of chemicals not produced/imported into the EEA have been made by countries such as Germany and Sweden. Such countries have consistently supported the implementation of stricter chemical regulations in Europe. This support has been based partly on a desire to export their stricter domestic chemical standards and policies to the European level, alongside efforts to further strengthen their domestic regulations (see e.g., Selin 2007).

Selin, H., 2007. Coalition politics and chemicals management in a regulatory ambitious Europe. *Global Environmental Politics*, 7(3), pp.63-93.

5. Lines 277-78 state that 'the production of almost half of the substances included on the Candidate List 278 had already ceased before the listing' (also in the abstract, line 37). As far as I can see, this can not be derived from the data. Isn't it just as well possible that the chemicals involved had never been produced in the EEA at all? If this were the case for a reasonable number of chemicals, this would further nuance the picture sketched in the paper (see previous point).

Rp: As previously mentioned, information on the number of countries with production/import within the EEA was collected from the REACH Registration dossiers. A country is reported as having production/import when there are active or inactive registrants for the substance. An inactive registrant has produced/imported the substance in the past but ceased production/import at the time of REACH registration. Given that the number of registrants includes inactive registrants, we can only say for certain that production/imports had ceased before REACH implementation.

Unfortunately, there is no dataset that allows us to confirm whether those chemicals have ever been produced in the EEA. However, anecdotal evidence reveals that, for some substances included on the Candidate List, there was a decreasing trend in their use long before inclusion on the Candidate List,

while other substances on the Candidate List have never been produced or imported into the EEA. Musk Xylene and DIHP provide two interesting examples:

Musk Xylene (EC:201-329-4, on Sunset-date 2014):

The use of musk xylene continued to decline through the 1990s, as fragrance manufacturers voluntarily switched to alternative fragrance compounds. For example, musk xylene has not been used in Japanese products (on a voluntary basis) since 1982, and the Association of the German Toiletries and Detergents Industry (IKW) recommended the replacement of musk xylene by another compound in 1993. Production of musk xylene in the European Union came to a halt and, by 2000 (the last year for which full data are available), imports to Europe were only 67 tonnes, with China as the most important source. The estimated 2008 usage of musk xylene in the European Union was 25 tonnes.

DIHP EC:276-090-2, On the candidate list since early 2020:

The compound has never been registered under REACH (see link above for the press release from ECHA). It is hard to find information on production etc. about the compound. However, INERIS ("The French National Institute for Industrial Environment and Risks," under the Ministry of the Environment) released the following information regarding the decision. "It should be noted that this substance has not been produced in Europe since 2010 and that it is unlikely to be imported into the European Union since 2011." (<https://substitution-phtalates.ineris.fr/en/news/dihp-substance-very-high-concern-svhc-according-european-reach-regulation>)

6. Finally, for a non-specialist reader (and as far as word count allows), some parts of the results section would benefit from more explicit explanation of what we actually see in the figures and tables (e.g Fig. 2, lines 207ff).

Rp: In the revised version of the paper, we do so. We have added notes to figures and tables, following the guidelines of the journal.

Reviewer #3:

1) Focus of the study: 1.1) The authors focus on authorisation as one of the two regulatory instruments of REACH (with restrictions being the second instrument). This is reasonable for the analysis but also bears a risk of ignoring that authorisation and restriction are not separate regulatory steps, but interconnected and complementary. In fact, while a restriction procedure can be initiated for any chemical (not just SVHC), member states can decide to start a restriction procedure on a chemical included on the candidate list if it appears that an authorisation procedure may not sufficiently cover the risk. For instance, this is the case if a SVHC is not widely produced in the EU, but imported in large quantities from outside the EU. A recent and prominent example is Dechlorane Plus, which was included on the Candidate list in 2018 due to its vPvB and potentially ED properties (see ECHA Candidate list table). This chemical is only produced outside the EU, still, it is highly economically relevant in the automotive and aviation industry. The Norwegian competent authorities, therefore, started a restriction procedure in 2020. The authors may refer to the restriction proposal on the ECHA website, and a corresponding Eftc report prepared under the Stockholm Convention, for further information. In summary, it is not correct that inclusion of a chemical on the Candidate list is a one-way road to authorisation. As a consequence, it is not justified to conclude that chemicals that enter the Candidate list which are not produced in the EU are necessarily 'of secondary importance'. 1.2) Note, further, that inclusion of a SVHC on the candidate list is just the first step for triggering authorisation. From the Candidate list, chemicals will be prioritised to be included on the Authorisation list'. Prioritisation occurs via a criteria-based selection process, using EU tonnage volume as an approximative criterion for both economic relevance and exposure (see https://echa.europa.eu/documents/10162/17232/recom_gen_approach_svhc_prior_2020_en.pdf/fbbd748b-22dc-38c2-9b4c-58c6bc80c930).

I therefore advise the authors to either better embed their results in the regulatory context of REACH, or consider to conduct their analysis for SVHC placed on the authorisation list only.

Rp: The reviewer makes several important points.

First, let us emphasize that our analysis does include imported amounts. Dechlorane Plus, which is the example raised by the reviewer, is included in our modelling as being on the Candidate List, with two importing countries and a tonnage band of 10-100 tonnes per annum. We have updated the description of our parameters ('Country count' and 'Tonnage Band') to make this clear.

Furthermore, we agree with the reviewer that our results can be put in a better regulatory context by more accurately describing REACH. We have therefore revised parts of the introduction and the discussion to emphasize that the Candidate List is not the only way to restrict the use of a compound.

To disentangle whether chemicals ended up on the Candidate List because they were first listed on the Restriction List, we investigate the overlap among the CL, the chemicals suggested for the Restriction List, the chemicals included on the Restriction List, and the full REACH registry. In addition, we compared the timing of listing on the Restriction List and the CL to see where chemicals were listed first. As seen in the figure below, out of 303 chemicals included on the Candidate List as of February 2020, 129 chemicals been also suggested or included on the Restriction List.

When analyzing the timing of the listing, we see that, for those chemicals added both on the Candidate and Restriction List, it is almost equally likely that a compound on the CL becomes restricted afterwards, as it is that a restricted compound becomes part of the CL. The same trend can be seen for the suggested restriction list, where the first suggestion for a restriction is about equally likely to come before or after the inclusion on the CL. However, given the scope and space limitations, we will refrain from including this information in the manuscript.

All in all, we have revised the manuscript to better describe the scope of our analysis, i.e., SVHCs restricted under the Authorization Program. We have, however, not changed the focus of the paper to include an analysis of the Restriction Program. We have also softened the language on some of the discussion points and made it clearer that the tonnage band and country count parameters include imports.

Specific changes have been made to address your comments.

Text moved from the Method section to the Introduction and modified to more explicitly describe the scope of the study.

“REACH consists of four complementary programs [7, 8]. Through the Registration program, companies are required to submit dossiers containing information about the properties and uses of chemicals. Through the Evaluation program, the European Chemical Agency (ECHA) checks some of these dossiers for compliance with the information requirements. The two key REACH programs that regulate chemical risk are Authorization and Restriction. Through the Authorization program, the manufacture and use of chemicals of substances of very high concern (SVHCs), which may have serious effects on human health and the environment, can be subjected to binding limitations and conditions, including complete prohibitions. The Restriction program also seeks to ensure that the risks from hazardous substances are properly controlled by prohibiting specific problematic uses of specific substances. However, unlike Authorization, Restriction is not limited to SVHCs. In this study, we investigate the listing of SVHCs under the Authorization program.”

L97: Changed phrasing to avoid the word restriction because this is a separate process

“Inclusion of a substance on the Candidate List is thus a first step toward requiring EC authorization for a compound’s manufacture, import and use [9]”

L257: Removed one sentence from the paragraph to soften the message somewhat.

The sentence that was removed stated

“This raises the question of whether the Candidate List prioritization process is contributing to overall hazard reduction. “

L362: We also acknowledge that our analysis is focused on the listing of SVHCs under the Authorization program, but hazardous substances also can be restricted under the Restriction program, which is beyond the scope of the study.

To clarify that imports were also considered in the study, we added the word at numerous locations throughout the text.

We also added the following clarification of the parameter.

456: “Information on the production/import tonnage and number of countries with production/import within the EEA was collected from the REACH Registration dossiers. A country is reported as having production/import when there are active or inactive registrants for the substance. An inactive registrant has produced/imported the substance in the past but ceased production/import at the time of REACH registration.

Finally, do also note that we see no significant difference between the annual produced/imported tonnages of the group compounds on the Candidate List versus the Authorization list. This could be explained because not all the substances recommended for inclusion on the Authorization List (i.e., prioritized) are finally included on the Authorization List.

2) Selection of toxicological variables: The current analysis uses variables referring to CMR properties and environmental properties, acute and chronic aquatic toxicity. While the human health related hazard categories match those also used for the selection of SVHC under REACH, the environmental indicators used in the study don't. The identification of a SVHC is based on hazard categories explained in REACH Article 57a-e. The corresponding identification criteria are listed in the Guidance on SVHC identification.

https://echa.europa.eu/documents/10162/2324906/svhc_en.pdf/8faef33c-b46e-4186-8b7c-8cfbeccd0812).

Specifically, the combination of persistence, bioaccumulation and/or toxicity (PBT), or the combination of high persistence and toxicity is crucial for SVHC classification because it points to long-lasting and potentially irreversible impacts to ecosystems. In the current analysis, however, environmental hazards are reflected by toxicity indicators ONLY. Therefore, and given the focus on the REACH Candidate list in comparison to other existing lists of harmful chemicals, the analysis is of little informational value. I recommend to re-do the analysis and explicitly consider the combination of P, B and/or T indicators to better capture the environmental concerns in the regression analysis. It may also be necessary to re-calculate the environmental score.

Rp: This is a misunderstanding – the environmental score used in the analysis includes both the B and P criteria. This is because the toxicity threshold needed to trigger labelling with GHS codes H400 and H410 is based on the compound’s B and P properties. For instance, classification of chronic toxicity category 1 (i.e., GHS410, Long-Term Aquatic Hazard; signal word = ‘warning’) depends on degradability

(P) and capacity for bioaccumulation (B). For algae, crustaceans or fish, the NOEC (i.e., the concentration at which no effects occur) is less than 0.01 mg/L when the compound is rapidly degradable. In contrast, the NOEC is less than 0.1 mg/L for a non-rapidly degradable compound. For a non-rapidly degradable compound that also bioaccumulates, the NOEC is less than 1 mg/L. See

https://echa.europa.eu/documents/10162/2324906/clp_en.pdf/58b5dc6d-ac2a-4910-9702-e9e1f5051cc5).

The P and B properties are thus integrated in the environmental score used in our analysis. To avoid further confusion, we have updated our description of this parameter and added a reference to the relevant guidance document.

“It should be noted here that the underlying GHS classifications for environmental hazard account for persistence (P) and bioaccumulation (B). The classification depends on the toxicity of the compound, where a compound is labelled as hazardous to the environment at different levels of toxicity depending on its P and B properties. For instance, a rapidly degradable compound which does not bioaccumulate will be classified as GHS410 (Long-Term Aquatic Hazard; signal word: warning) if effects can be seen at concentration below 0.01 mg/L for algae, crustaceans, or fish. If the compound is not rapidly degradable, the classification applies if effects are seen at concentrations below 0.1 mg/L, and if the compound is also bioaccumulative the classification applies if effects are seen at concentrations below 1 mg/L [32]. Thus, less toxicity is required for the classification if the compound is also P and B.”

We would also like to point out that we did indeed try to gather the PBT assessments from the REACH registry at the onset of the project. However, information on PBT-assessment is missing in most of the registration dossiers.

3) Scoring of human and environmental hazard variables: The authors explain in line 431 onwards that human health and environmental hazard indicators were transformed into scores. I fail to understand the method for determining the score. The authors should provide a more transparent description of their approach, maybe in the Supplementary Materials section of the paper. Furthermore, I miss a list of score values for the chemicals included in the analysis under each list.

Rp: We agree with the reviewer that the scoring was not presented transparently enough. We have added the following sentence to the manuscript:

“A CMR score was devised where the percentage of all firms reporting GHS codes for C, M and R properties were summed and the difference in severity between the signaling words was accounted for by multiplying all percentages associated with ‘danger’ by two. Finally, the score was normalized to vary between 0 and 1.”

Moreover, note that all material used in the study is now uploaded to a repository. In the repository, you can find information on the scores for all chemicals included in the analysis under each list.

DOI for data plus code repository: 10.5281/zenodo.6563557

4) Comparison of results across lists: It is interesting to compare the determinants across different lists of SVHC in Europe. But the inclusion criteria for SVHC may differ between the Candidate list, SIN and PRIO. The authors should explain these criteria.

Rp: We agree with the reviewer that this could have been made clearer. Please see our answer to a similar comment from reviewer 1.

5) Documentation of data: Maybe I overlooked it but I failed to find a documentation of all data used for the analysis (in particular the list of chemicals used, and the corresponding values of the variables). To make the analysis traceable and transparent please add this information (either as part of the supplementary material, or as a separate online data repository).

Rp: We have added a full S.I. and links to repositories containing all data and code needed to perform all modeling steps in the manuscript.

6) Additional suggestions:

6.1: Line 2/3: Given the points explained above I would suggest toning down and re-formulate the title of the paper. The results cannot be interpreted as 'determinants of the regulation of SVHC in Europe' considering that the Candidate list is just one step in the selection process and authorization is one of two complementary instruments for regulatory control.

Rp: As previously described, we have toned down some of the statements in the paper following the reviewers' suggestions. We also have clarified that our study concerns SVHCs regulated under the Authorization program. We do not want to change the title of the paper. We still feel that it is a good representation of the content, given the clarifications and delimitations. Specifically, we believe the title is appropriate because the abstract makes it clear that our focus is on the Candidate List.

6.2: Lines 77/82: It is too vague to describe PBT/vPvB properties as posing 'broad environmental concerns'. To the contrary, PBT/vPvB denote hazard categories indicating composite hazards and, thus damage to ecosystems over time. The formulation should become more specific.

Rp: We have tried to refrain from using specific numbers on logKow values, biodegradation rates, etc., to keep the manuscript readable for a broader audience. However, we agree with the reviewer that the sentence in question was too vague. The paragraph now reads.

"REACH aims to reduce hazards to both public health and the environment [7, 8]. Thus, criteria for identifying substances as SVHCs are whether they are carcinogenic, mutagenic, or toxic for reproduction (CMR); persistent, bioaccumulative and toxic (PBT); very persistent and very bioaccumulative (vPvB); or raise equivalent levels of concern [9]. CMR substances are directly hazardous to human health, while PBT and vPvB substances pose long-term, unpredictable risks due to their longevity, irreversible nature, and tendency to accumulate in the food chain."

6.3: Lines 261-278 and lines 348-353: I suggest toning down these text passages. Looking at the result, one may conclude that 'absence from production in Europe' is the 'most significant variable' determining inclusion on the Candidate list. Considering that chemicals listed under the Candidate list can also be selected for restriction, the conclusion that the Candidate list is systematically biased towards chemicals which are economically not relevant in the EU is not warranted. See also the arguments provided under 1).

Rp: We agree with the reviewer that these statements were too strongly phrased, given that we did not clarify that we include imports in our Country Count and Tonnage Band parameters. In the Method section, we have now clarified that these economic parameters include imports. We also clarified that

we limit the scope of this study to the Candidate List and that restrictions are an additional available mechanism beyond the scope of this study.

6.4 Line 385/386: Though restriction is not limited to SVHC, past years' restriction procedures have often focused on SVHCs. Moreover, several restrictions were initiated following-up on authorization procedures, e.g. because the Commission concluded that there is a risk of a continued use of the chemical for the EU as a whole, or because a refused authorization does not prevent imports of a SVHC into the EU.

Rp: As mentioned before, we have clarified that the Candidate List is a first step, but not the only step towards inclusion on the Authorization List.

6.5 Line 467: Please explain how scientific journals were selected.

Rp: As now noted in the article, "The journals were chosen to 1) provide representation in the fields of toxicology and ecotoxicology, 2) publish mainly original research, 3) have an international focus and a peer-review principle and 4) have overall high scientific standards (thus excluding any potentially predatory journals) (Supplementary Table 1). These criteria were manually assessed based on information from the journals' homepages and bibliometric information from the Scopus and Web of Science databases."

With this journal selection process, we have aimed to get a representative sample of the fields, rather than capture everything and dilute the "knowledge" signal with predatory or low-impact journals.

Reviewers' Comments:

Reviewer #1:

Remarks to the Author:

I read through the authors' responses to the 3 reviewer comments and also re-read the manuscript. The authors responded adequately, in my view, to the comments and I would recommend you accept the revised manuscript.

Reviewer #2:

Remarks to the Author:

The authors have made a serious and, as far as I am concerned, successful effort to address my earlier concerns. As stated before, given my own expertise, I trust that the statistical 'technicalities' of the analysis have been covered by other reviewers. Having said that, I recommend publication of the paper in its present form.

Reviewer #3:

Remarks to the Author:

Many thanks for sending the revised paper.

I appreciate the revisions made so far. Still, I have two main concerns which I would ask the authors to address before I can recommend the paper for publication in Nature Communications.

1) The conclusions conducted are not properly backed-up by the setup of the analysis. The analysis focuses on the chemicals included in the Candidate List. While the reasons provided by the authors for using the Candidate List chemicals are understandable from a technical perspective (line 236-240), I do not consider it justified to claim that results inform about the 'drivers for the regulation of SVHC under REACH'. In fact, results of their analysis inform about the drivers for including SVHC on the Candidate List, which is the superset of SVHC from which chemicals subject to authorisation, i.e. to which regulatory action factually applies, are selected. But it does not per se inform about the drivers for regulation. To this end, I also think that the title of the paper raises misleading expectations. As a matter of transparency and integrity, and since the authors know the 86 chemicals on the Authorisation List, they should, in addition to the analysis based on the Candidate List, check and explicitly discuss in the paper whether their findings are confirmed by the SVHC on the Authorisation List.

2) The authors make a number of statements which are in my view unnecessarily unbalanced. The paper could be strengthened, considering also the audience of the journal, by presenting arguments more nuanced.

-Line 87 (and also line 151/152) "REACH aims to reduce hazards to both public health and the environment". I suggest to quote the aims of REACH directly, as formulated in the REACH legislation, Article 1: "The purpose of this Regulation is to ensure a high level of protection of human health and the environment, including the promotion of alternative methods for assessment of hazards of substances, as well as the free circulation of substances on the internal market while enhancing competitiveness and innovation." Thus, besides a high level of protection REACH also intends to support EU chemical industry. It is obvious that this may create trade-offs, which the authors address with their analysis, but it is simply not correct to emphasize the aim of health and environmental protection ONLY. And it may even be seen as an argument explaining why the list of SVHC is also driven by economic interests. I would encourage the authors to discuss this in a more nuanced way.

-Line 133 "The multinational character of REACH implies that the structure and economic importance of the domestic chemicals industry might also be of considerable importance for the listing of SVHCs [...]" The authors cite two scientific publications which do neither address nor support the authors' argument that multinationality (of REACH) as such reduces the effectiveness of regulation. While the

argument seems plausible it should be underpinned by appropriate scientific studies/evidence.

-Line 143 "Policy makers have been more likely not to regulate something that was later found to be harmful than to err on the side of caution. The REACH listing procedures of SVHCs might enhance this tendency, as the consultation process provides producers with an opportunity for exposing weaknesses and uncertainties in the scientific case for inclusion of chemical substances on the Candidate List." Just as a point for reflection: this could also be regarded as a fundamental principle of democracy in the decision-making process, considering that besides producers everyone is invited to provide comments during the consultation phase. It is indeed worrying that some stakeholders seem to be more active during this phase than others. If producers are also more successful in promoting their interests it is not necessarily a shortcoming of the institutional design alone, but probably also a failure of responsible decision-making on the side of policy makers.

-Line 314 "Regardless of the explanation, the evidence raises concerns on the actual extent to which the Candidate List has encouraged manufacturers to remove substances of very high concern from the market, when the production/import of almost half of the substances included on the Candidate List had either already ceased before the listing, or the compound has never been produced or imported in the EEA at all." Again, it would be important if this also holds for those SVHC which are on the Authorisation List, which is the relevant list for taking regulatory action. If not, I think the authors' conclusion is not valid.

-Line 387 "However, the most important and significant variable explaining the odds of inclusion on the Candidate List is not how dangerous a chemical is but the fact that it is neither produced in Europe nor imported into Europe. The lack of production and import of the chemicals in the European Economic Area is the most important determinant for inclusion on the list." This would be an interesting result if these findings also hold for the SVHC on the Authorisation List. Since the Candidate List is not the list triggering regulatory action, it is probably not even surprising that multiple factors determine the selection of chemicals.

Finally, I would suggest some minor corrections:

-Line 97/98: It should be mentioned that substances placed on the Authorisation List can no longer be produced or marketed after a defined 'sunset date', so the prohibition to produce and market is not adopted immediately.

-Line 159/160: It should be "Third, since it is well known..."

REVIEWER COMMENTS

Reviewer #3

- 1) The conclusions conducted are not properly backed-up by the setup of the analysis. The analysis focuses on the chemicals included in the Candidate List. While the reasons provided by the authors for using the Candidate List chemicals are understandable from a technical perspective (line 236-240), I do not consider it justified to claim that results inform about the 'drivers for the regulation of SVHC under REACH'. In fact, results of their analysis inform about the drivers for including SVHC on the Candidate List, which is the superset of SVHC from which chemicals subject to authorisation, i.e. to which regulatory action factually applies, are selected. But it does not per se inform about the drivers for regulation. To this end, I also think that the title of the paper raises misleading expectations. As a matter of transparency and integrity, and since the authors know the 86 chemicals on the Authorisation List, they should, in addition to the analysis based on the Candidate List, check and explicitly discuss in the paper whether their findings are confirmed by the SVHC on the Authorisation List.

Rp: We have followed the reviewer's suggestions and performed the analysis for the chemicals on the Authorization List. The results of such regressions are presented in the Supplementary Material (See Tables S5 and S6 as well as Figures 5 and 6). Even though the statistical power is lower for this list than for the Candidate List, they do confirm the findings of the paper.

We added the Section "What drives the inclusion on the Authorization List?" explaining that such analysis was performed, and that the findings for the Authorization List support our previous findings, i.e., the lack of production and import of the chemicals in the European Economic Area is also the most important determinant for inclusion on the Authorization List (see lines 359-396). Due to space constraints, we refer the interested reader to the Supplementary Material for full details.

Based on the results of this additional analysis, we see no reason to change the title of the paper.

- 2) The authors make a number of statements which are in my view unnecessarily unbalanced. The paper could be strengthened, considering also the audience of the journal, by presenting arguments more nuanced.

-Line 87 (and also line 151/152) "REACH aims to reduce hazards to both public health and the environment". I suggest to quote the aims of REACH directly, as formulated in the REACH legislation, Article 1: "The purpose of this Regulation is to ensure a high level of protection of human health and the environment, including the promotion of alternative methods for assessment of hazards of substances, as well as the free circulation of substances on the internal market while enhancing competitiveness and innovation." Thus, besides a high level of protection REACH also intends to support EU chemical industry. It is obvious that this may create trade-offs, which the authors address with their analysis, but it is simply not correct to emphasize the aim of health and environmental protection ONLY. And it may even be

seen as an argument explaining why the list of SVHC is also driven by economic interests. I would encourage the authors to discuss this in a more nuanced way.

Rp: We agree and have revised the sentence starting at Line 87 following the reviewer's suggestion:

"The main aims of REACH are to ensure a high level of protection for human health and the environment, including the promotion of alternative test methods, as well as the free circulation of substances on the internal market and the enhancement of competitiveness and innovation [7, 8,9]."

We have also revised Line 151/152:

"Our choice of variables is based on the criteria for identifying substances as SVHCs, and on a review of relevant literature"

-Line 133 "The multinational character of REACH implies that the structure and economic importance of the domestic chemicals industry might also be of considerable importance for the listing of SVHCs [...]" The authors cite two scientific publications which do neither address nor support the authors' argument that multinationality (of REACH) as such reduces the effectiveness of regulation. While the argument seems plausible it should be underpinned by appropriate scientific studies/evidence.

Rp: We realize that the paragraph was not clear since we did not intend say that the multinationality of REACH reduces the effectiveness of regulation. We intended to say

"The multinational character of REACH implies that the structure and economic importance of the domestic chemicals industry might also be of considerable importance for the listing of SVHCs",

and here we mean that, since each country is expected to support/protect its national chemical industry, it might be more difficult to regulate chemicals produce by many EU countries since they affect the economic interests of many actors. The references we cite present evidence that it is more difficult to regulate when there are many vested interests.

To avoid confusion, we rephased the sentence as

"The structure and economic importance of the domestic chemicals industry might also be of considerable importance for the listing of SVHCs". Specifically, chemicals produced or used in large quantities, or by many EU countries, might be more difficult to regulate because they affect the economic interests of many actors, and the likelihood of successfully lobbying policymakers increases with the number of interest groups pushing for the same policy goal [19, 20].

We also revised on the references cited so as the evidence present comes so close as possible to the statement.

[19] Klüver, H. 2011. The contextual nature of lobbying: Explaining lobbying success in the European Union. *European Union Politics*, 12(4), 483-506.

[20] Eising, R. 2007. The access of business interests to EU institutions: towards elite pluralism? *Journal of European Public Policy*, 14(3), 384-403.

-Line 143 "Policy makers have been more likely not to regulate something that was later found to be harmful than to err on the side of caution. The REACH listing procedures of SVHCs might enhance this tendency, as the consultation process provides producers with an opportunity for exposing weaknesses and uncertainties in the scientific case for inclusion of chemical substances on the Candidate List." Just as a point for reflection: this could also be regarded as a fundamental principle of democracy in the decision-making process, considering that besides producers everyone is invited to provide comments during the consultation phase. It is indeed worrying that some stakeholders seem to be more active during this phase than others. If producers are also more successful in promoting their interests it is not necessarily a shortcoming of the institutional design alone, but probably also a failure of responsible decision-making on the side of policy makers.

Rp: We appreciate your reflection and agree with it. business and industry are in general in a better position to represent their interests and achieve their preferences than organizations representing diffuse interests. We reflect on this on the conclusions (see lines 451-461) and argue that accounting for such asymmetries must require a redesign of the process, but the analysis of the redesigns that should be implemented goes beyond the scope of this paper.

-Line 314 "Regardless of the explanation, the evidence raises concerns on the actual extent to which the Candidate List has encouraged manufacturers to remove substances of very high concern from the market, when the production/import of almost half of the substances included on the Candidate List had either already ceased before the listing, or the compound has never been produced or imported in the EEA at all." Again, it would be important if this also holds for those SVHC which are on the Authorisation List, which is the relevant list for taking regulatory action. If not, I think the authors' conclusion is not valid.

Rp: As described in our response to your first comment, our results do indeed hold for those SVHC which are on the Authorisation List. This has, as previously described, been clearly stated in the paper (see lines 359-396).

-Line 387 "However, the most important and significant variable explaining the odds of inclusion on the Candidate List is not how dangerous a chemical is but the fact that it is neither produced in Europe nor imported into Europe. The lack of production and import of the chemicals in the European Economic Area is the most important determinant for inclusion on the list." This would be an interesting result if these findings also hold for the SVHC on the Authorisation List. Since the Candidate List is not the list triggering regulatory action, it is probably not even surprising that multiple factors determine the selection of chemicals.

Rp: We agree with the reviewer that the Candidate List is not the list triggering regulatory action. The Candidate List is, however, a first step toward authorization, and as shown by the statistics in Table 1, there are not statistically significant differences on the variables under analysis for the substances on the Candidate and Authorization Lists. As shown by the analysis of the drivers of inclusion on the Authorization List, the lack of production and import of the chemicals in the European Economic Area is also the most important determinant for inclusion on the Authorization List (see lines 359-396).

We do not deem it surprising that multiple factors determine the selection of chemicals. But the clear salience of the lack of production and import of the chemicals in the European Economic Area seems surprising to us. Based on the analysis and evidence presented, it is our view that the actual effect of the Candidate List in removing SVHCs from the market can be called into question. We are grateful for the reviewer's efforts and constructive suggestions. We believe that they have helped us to significantly improve the paper.

Minor corrections:

-Line 97/98: It should be mentioned that substances placed on the Authorisation List can no longer be produced or marketed after a defined 'sunset date', so the prohibition to produce and market is not adopted immediately.

Rp: In the revised version of the paper, we have corrected this. The line reads now as:

In turn, substances placed on the Authorization List cannot be made available on the market after a defined sunset date, unless the EC grants an authorization.

-Line 159/160: It should be "Third, since it is well known..."

Rp: We have corrected this, thanks.

Reviewers' Comments:

Reviewer #3:

None